# Predicting the probability of death using proteomics

Thjodbjorg Eiriksdottir [1], Steinthor Ardal[1], Benedikt A. Jonsson[1], Sigrun H. Lund [1], Erna V. Ivarsdottir [1], Kristjan Norland[1], Egil Ferkingstad [1], Hreinn Stefansson[1], Ingileif Jonsdottir [1,2,3], Hilma Holm [1], Thorunn Rafnar [1], Jona Saemundsdottir[1], Gudmundur L. Norddahl[1], Gudmundur Thorgeirsson[1,2,3], Daniel F. Gudbjartsson [1,2], Patrick Sulem [1], Unnur Thorsteinsdottir[1,2], Kari Stefansson [1,2✉] & Magnus O. Ulfarsson [1,2✉]

Predicting all-cause mortality risk is challenging and requires extensive medical data. Recently, large-scale proteomics datasets have proven useful for predicting health-related outcomes. Here, we use measurements of levels of 4,684 plasma proteins in 22,913 Icelanders to develop all-cause mortality predictors both for short- and long-term risk. The participants were 18-101 years old with a mean follow up of 13.7 (sd. 4.7) years. During the study period, 7,061 participants died. Our proposed predictor outperformed, in survival prediction, a predictor based on conventional mortality risk factors. We could identify the 5% at highest risk in a group of 60-80 years old, where 88% died within ten years and 5% at the lowest risk where only 1% died. Furthermore, the predicted risk of death correlates with measures of frailty in an independent dataset. Our results show that the plasma proteome can be used to assess general health and estimate the risk of death.

[1] deCODE Genetics/Amgen, Inc., Reykjavik, Iceland. [2] University of Iceland, Reykjavik, Iceland. [3] Landspitali, The National University Hospital of Iceland, Reykjavik, Iceland. ✉email: kstefans@decode.is; mou@hi.is

The ability to predict when someone will die is not something you would wish upon yourself or your friends. It could, however, prove useful in the delivery of healthcare and biomedical research. It is often possible to give a meaningful prediction of how long individuals with specific diagnoses will live[1], but predicting when an individual will die from any cause is altogether a different matter.

Several diseases, lifestyle[2–4], social and psychological factors[5] associate with all-cause mortality. Commonly used risk factors for all-cause mortality are age, sex, traditional cardiovascular risk factors such as systolic blood pressure, cholesterol levels, smoking, and diabetes, cardiovascular disease, cancer, alcohol consumption, body mass index (BMI), and creatinine levels[6–8]. Among other biomarkers of all-cause mortality are brain age estimated from structural magnetic resonance images[9], DNA methylation[10], and telomere length[11]. Recently, circulating metabolic biomarkers have been found to associate with the risk of all-cause mortality. In a study of 44,168 individuals, where 5512 died during follow-up, 14 metabolic biomarkers were found to improve 5 and 10-year all-cause mortality predictions over conventional risk factors[8]. Another study of 17,345 participants identified 106 metabolic biomarkers that improved short-term all-cause mortality risk prediction over established risk factors[7]. In a study of 3523 participants from the Framingham Heart Study, 38 of 85 preselected circulating protein biomarkers associated with all-cause mortality and improved all-cause mortality prediction over cardiovascular risk factors[12]. Similarly, 56 peptides (31 proteins) correlated with 5-year mortality in a study of 2473 older men. A panel of those peptides improved the predictive value of a commonly used clinical predictor of mortality[13].

With the advent of new technology such as SOMAmers[14] or proximity extension assays[15], it is possible to simultaneously measure levels of thousands of proteins efficiently. Several studies using this technology have shown the plasma proteome to be heavily associated with age and life span[16–20]. A study of 997 participants associated 651 out of 1301 proteins with age, found that a 76-protein proteomic age signature associated with all-cause mortality independent of chronological age, and created a seven-protein mortality predictor[18]. In a study of 1025 older adults, 754 of 4265 proteins were associated with age. A proteomic age model using the age-associated proteins predicted mortality better than chronological age[19]. Another study of 4263 participants measured 2925 proteins to evaluate how circulating protein profile changes over the life span[20]. Some studies have used large proteomics datasets to predict other health-related factors. A protein-based risk score for cardiovascular outcomes in a high-risk group was developed using 1130 candidate plasma proteins[21]. In addition, ~5000 plasma proteins were used to predict health states, behavior, and incident diseases, with performance comparable to traditional risk factors, in 16,894 participants[22]. These studies underscore the value of using plasma levels of a large number of proteins to search for biomarkers in health and diseases.

Here we apply plasma levels of 4684 proteins determined with SOMAmers to predict both long- and short-term all-cause mortality. We developed and tested predictors using a dataset of 22,913 individuals, of whom 7061 died during the study period. Predictors using proteins were compared to predictors using only conventional risk factors, and we examined prediction performance for various causes of death. We also explored how individual proteins associate with all-cause mortality and various causes of death. Using an independent dataset of 8814 individuals, we correlated the predictor with several frailty measures and known risk factors of mortality.

## Results
The data came from four sources; the Icelandic cancer project (ICP)[23], deCODE health study (dHS)[24], and various smaller projects from two distinct time periods (VSP1 and VSP2). The ICP and VSP1 data, used for model development, included 22,913 participants aged 18–101 (mean 56.6, sd. 17.4) sampled between the years 2000-2006 (Supplementary Fig. 1), of whom 10,136 were 60 years old or older (mean 73.0, sd. 7.8). The average follow-up time for this group was 13.7 (sd. 4.7) years, until death or the end of the study period at the end of 2018. At the time of sample collection, 7.0% had coronary artery disease (CAD), 5.2% history of myocardial infarction (MI), 2.5% history of stroke, and 23.9% had been diagnosed with cancer. Since most of this dataset was collected for cancer research, it has about three times higher cancer prevalence than the more recently collected dHS sample set. During the study period, 7061 participants (30.8%) died at an average age of 81.2 (sd. 10.7) years. Of those who died, 38.1% of deaths related to neoplasms, 8.4% to the nervous system, 35.0% to the circulatory system, 7.7% to the respiratory system, and 10.8% to other internal causes. Table 1 lists the baseline characteristics for all datasets. For every participant, 4905 protein measurements (aptamers) measuring levels of 4684 different proteins in plasma were available after a quality check. Models were developed using 70% of the data, and results were reported on the remaining 30%.

**Prediction performance at different time points**. In Fig. 1, we demonstrate the discriminatory power of our all-cause mortality prediction models by using a receiver operating characteristic (ROC) curve and the area under the curve (AUC) both for all participants (Fig. 1a, b) and restricted to 60 years or older (Fig. 1c, d).

The AUC for all participants using only age and sex (age and sex model) increases with the time from sample collection (Fig. 1a). Adding disease and lifestyle variables to the model (baseline model) increases the AUC over the age and sex model. Adding the growth/differentiation factor 15 (GDF15) protein measurement, which has the strongest association with all-cause mortality of all 4905 protein measurements, to the age and sex model (GDF15 model) yielded a better predictor than the baseline model. Adding more protein measurements to the age and sex model gave an even better prediction model (protein model). The difference between the four prediction models was greater for short-term predictions than long-term predictions.

The proteins in the protein model were chosen separately for predictions of all-cause mortality within 1,2,…,15 years. In general, the short-term predictions needed fewer proteins than the long-term predictions. For instance, the Boruta[25] feature selection chose 209 protein measurements for prediction of death within 1 year, but 454 protein measurements for death within 15 years. 135 protein measurements were constantly chosen for death within each of 1,2,…,15 years. The L1[26] penalty reduced the model to 81 protein measurements for prediction of death within 1 year and 192 for death within 15 years, but the biggest model, which was for prediction of death within 13 years, used 219 protein measurements. Ten protein measurements were chosen in every model. The 5-year predictor included 117 protein measurements, the 10-year predictor included 199, and the 2-year predictor included 98 protein measurements (Supplementary Table 1). The features and coefficients of the 5-year predictor are in Supplementary Data 1.

As an example of short-, intermediate-, and long-term predictions, we looked at the prediction of death within 2, 5, and 10 years (Supplementary Table 2).

Figure 1b depicts the ROC curves for all-cause mortality within 5 years for all participants. The AUC for the age and sex model was 0.852. The baseline model had an AUC of 0.885, an increase of 0.033 ($p = 2.4e{-}11$) over the age and sex model. The GDF15 model had an AUC of 0.893, with an increase of 0.008 ($p = 8.7e{-}2$) over

| Table 1 Characteristics of all study participants by age and sample sets. | | | | | | |
|---|---|---|---|---|---|
| Characteristic | ICP + VSP1 All N = 22,913 | ICP + VSP1 60+ N = 10,136 | dHS All N = 8814 | dHS 60+ N = 3684 | VSP2 All N = 6798 | VSP2 60+ N = 2611 |
| Men | 9991(43.6) | 4816(47.5) | 3876(44.0) | 1652(44.8) | 2657(39.1) | 1109(42.5) |
| Women | 12,922(56.4) | 5320(52.5) | 4938(56.0) | 2032(55.2) | 4141(60.9) | 1502(57.5) |
| Follow up | 13.7(4.7) | 11.1(5.5) | 1.3(0.7) | 1.3(0.7) | 1.6(1.4) | 1.6(1.4) |
| Age | 56.6 (17.4) | 73.0(7.8) | 55.4(14.7) | 69.1(6.4) | 52.9(16.6) | 70.0(6.7) |
| Age-span | 18–101 | 60–101 | 18–96 | 60–96 | 18–98 | 60–98 |
| BMI | 26.5(4.6) | 26.6(4.4) | 28.6(5.3) | 28.9(5.1) | 27.8(5.4) | 27.7(4.9) |
| T2D | 969(4.2) | 809(8.0) | 437(5.0) | 299(8.1) | 258(3.8) | 177(6.8) |
| Statin use estimate | 1897(8.3) | 1617(16.0) | 1623(18.4) | 1256(34.1) | 1489(21.9) | 1140(43.7) |
| HT medication use | 7676(33.5) | 5599(55.2) | 4137(46.9) | 2553(69.3) | 3292(48.4) | 1962(75.1) |
| Smoker estimate | 2998(13.1) | 792(7.8) | 844(9.6) | 310(8.4) | 621(9.1) | 235(9.0) |
| CAD | 1608(7.0) | 1490(14.7) | 645(7.3) | 570(15.5) | 869(12.8) | 723(27.7) |
| History of MI | 1199(5.2) | 1095(10.8) | 249(2.8) | 202(5.5) | 394(5.8) | 314(12.0) |
| History of Stroke | 568(2.5) | 511(5.0) | 168(1.9) | 129(3.5) | 184(2.7) | 128(4.9) |
| Cancer diagnosis | 5484(23.9) | 3880(38.3) | 675(7.7) | 512(13.9) | 526(7.7) | 400(15.3) |
| Deaths | 7061(30.8) | 6222(61.4) | 25(0.3) | 22(0.6) | 83(1.2) | 74(2.8) |
| Age at death | 81.2(10.7) | 84.0(7.5) | 76.8(10.1) | 79.8(6.6) | 75.1(10.4) | 77.3(8.5) |
| *Cause of death* | | | | | | |
| Neoplasms | 2687(38.1) | 2098(33.7) | 12(48.0) | 11(50.0) | 49(59.0) | 43(58.1) |
| Nervous system | 596(8.4) | 550(8.8) | 1(4.0) | 1(4.5) | 3(3.6) | 3(4.1) |
| Circulatory system | 2472(35.0) | 2345(37.7) | 9(36.0) | 7(31.8) | 23(27.7) | 20(27.0) |
| Respiratory system | 544(7.7) | 507(8.1) | 0(0.0) | 0(0.0) | 6(7.2) | 6(8.1) |
| Other causes | 762(10.8) | 722(11.6) | 3(12.0) | 3(13.6) | 2(2.4) | 2(2.7) |

The numbers are number (percent of participants), number (percent of total deaths), mean (sd), or range

the baseline model, while the protein model yielded an AUC of 0.915, improving the baseline AUC by 0.030 ($p = 1.4e{-}9$).

Restricting the analysis to participants 60 years or older lowers the AUCs compared to models including all the participants. However, the differences from the baseline model were greater (Fig. 1c, d). The lower AUC and bigger AUC differences probably result from the exclusion of many easily classified participants younger than 60. For example, the youngest age group is at very low mortality risk and easily distinguished using the age variable. That is, the smaller age range reduces the importance of age. For the 5-year prediction, the AUCs were 0.750, 0.801, 0.820, and 0.853 for the age and sex, baseline, GDF15, and protein model, respectively. The differences from the baseline were −0.050 ($p = 4.3e{-}10$), 0.019 ($p = 3.4e{-}2$), and 0.053 ($p = 4.5e{-}9$) for the age and sex, GDF15, and protein models, respectively.

For 5-year prediction for all participants, the integrated discrimination improvement (IDI) for the protein model vs. the baseline was 0.115 (95% CI: 0.095–0.137). For older than 60 participants the IDI was 0.113 (95% CI: 0.092–0.136) for 5-year prediction. (Supplementary Fig. 2 and Supplementary Table 2).

When the predictors were applied to a subset of participants not diagnosed with any of the major diseases used in the baseline at the time of plasma collection, the protein model was still the best prediction model (Supplementary Fig. 3a). The baseline model and the GDF15 model still did better than the age and sex model, but the difference is much smaller than when the whole dataset is used. This is not surprising since information about the excluded diseases is essential to the baseline model. We also examined the discrimination power in participants 80 years or older separately, using the models trained for participants older than 60. The protein predictors discriminated better than the baseline model at every time point (Supplementary Fig. 3b).

Adding the baseline model variables to the GDF15 model and the protein model improved predictions for all time points, both for all and older than 60 participants (Supplementary Fig. 4 and Supplementary Table 3). However, we were more interested in what the protein measurements could do without information on

lifestyle and diseases; therefore, we did not include the baseline variables in the GDF15 and protein models. Excluding age and sex from the protein model reduced prediction performance slightly, especially for long-term predictions, but age and sex are an essential part of the GDF15 model. Since age and sex are easily obtainable features, we saw no advantage in excluding them from the models.

GDF15 is a powerful predictor on its own and as a part of the protein model. To examine what a protein model without GDF15 could do, we created a new protein model where we excluded GDF15 (Supplementary Fig. 4). There was no significant difference in the AUC between the protein model and the new protein model excluding GDF15. To see if other single proteins were good mortality predictors, we also tried models using age, sex, and WAP four-disulfide core domain protein 2 (WFDC2), Thrombospondin-2 (THBS2), or Anthrax toxin receptor 2 (ANTXR2). These were the proteins in addition to GDF15 with the strongest association with 5-year mortality and were all useful in predicting mortality. However, the GDF15 model remains the only single protein model to surpass the baseline model in prediction performance (Supplementary Fig. 5). Therefore, GDF15 cannot easily be swapped for any single candidate protein, but a combination of proteins can make up for performance loss from excluding GDF15.

**Other protein predictors**. Other protein-based mortality predictors have been developed. We tried to replicate them in our data and compared them to our predictors as shown in Supplementary Fig. 6. The difference between protein-derived age and chronological age, sometimes called predicted age difference (PAD), has been shown to be predictive of mortality[18,19]. We calculated a PAD and used it as a feature in a mortality prediction model. The PAD was predictive of mortality but was far from being as good of a predictor as GDF15 alone. We also tried a mortality predictor using the seven proteins shown to be useful mortality predictors by Tanaka et al.[18]. The seven-protein model

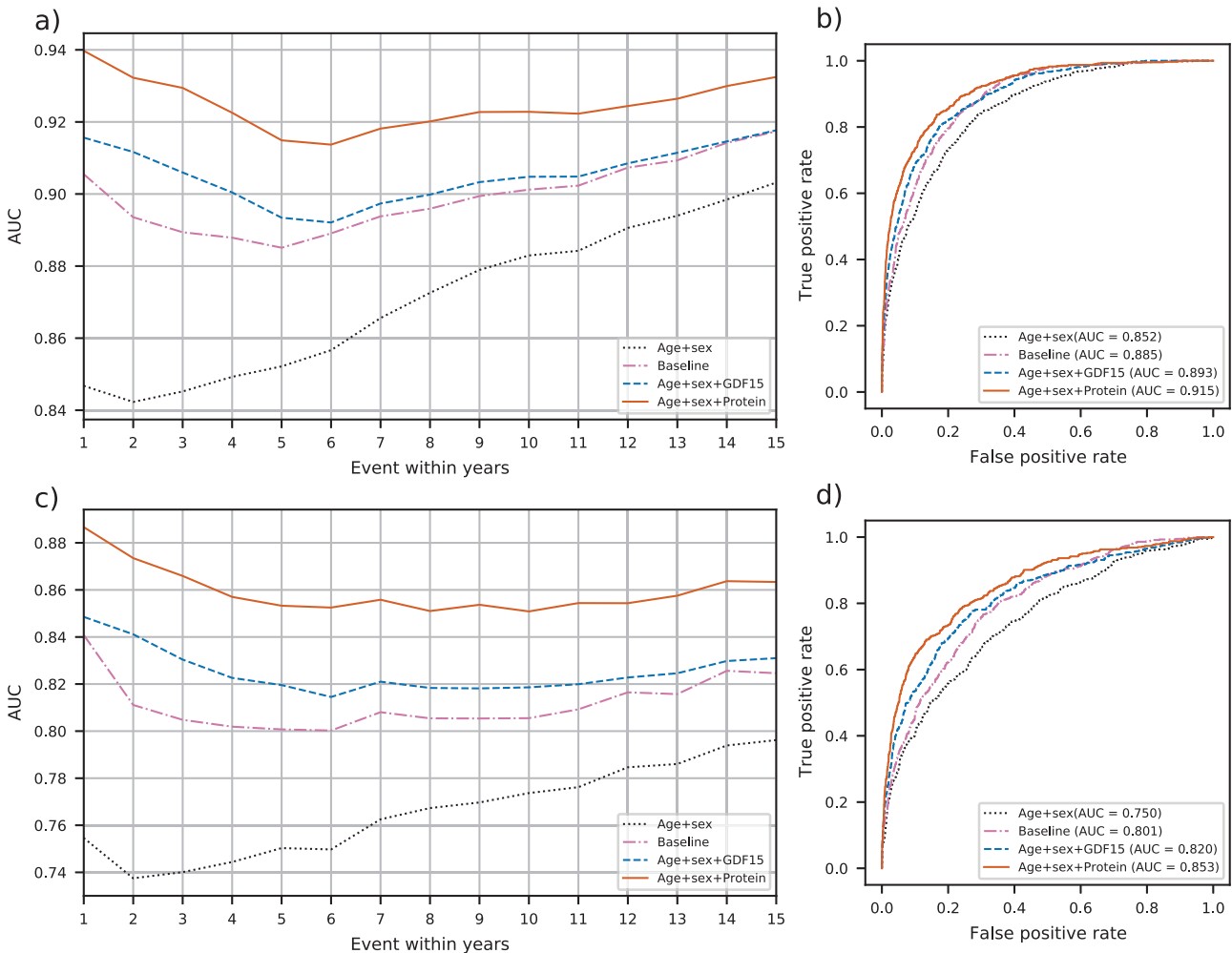

**Fig. 1 Discrimination power of different models for death within 1,2,...,15 years. a** AUCs for all participants, N = 6893. **b** ROC curves for death within 5 years for all participants. **c** AUCs for participants older than 60, N = 3052. **d** ROC curves for death within 5 years for participants older than 60.

performed better than our GDF15 model but did not reach our protein model performance. This is expected since GDF15 was one of the seven proteins and because our protein model makes use of more proteins. Finally, we tried using ten of twelve proteins used in a multivariable model by Ho et al.[12], adjusted for many of our baseline variables. This model also included GDF15 and had prediction performance between our protein model and the seven protein model which might be expected since it used more proteins than seven and fewer than our protein model. These proteins were selected out of a set of proteins targeted because of their high value for cardiovascular disease.

**Kaplan–Meier analysis.** We looked at Kaplan–Meier survival curves for participants in the ICP + VSP1 test set between 60 and 80 years old to reduce the effect of age. That included 2488 participants with mean age 70.1 (sd. 5.5), of whom 1312 (52.7%) died during the study period, 305 (12.3%) within 5 years, and 701 (28.2%) within 10 years from sample collection. The curves were plotted separately for the four prediction models. By splitting the Kaplan–Meier curves by quantiles of predicted 10-year risk, the proteins' discriminative power becomes evident (Fig. 2, Supplementary Fig. 7). Of the 5% (124 participants) predicted at the highest risk by the age and sex, baseline, GDF15, and protein model, 25%, 40%, 55%, and 67% died within 5 years, and 56%, 65%, 74%, and 88% within 10 years. Of the 5% (125 participants)

predicted at the lowest risk by each model, 8%, 5%, 8%, and 1% died within 10 years.

The protein model 5% high-risk group is younger and of more varied age (mean 74.0, sd. 4.9) than the baseline model group (mean 76.9, sd. 2.6). Likewise, the protein model 5% low-risk group is older and of more varied age (mean 62.7, sd. 2.4) than the baseline model group (mean 61.9, sd. 1.3).

In a group of over 80-year-old participants, we examined survival curves split by predicted ten-year risk. The 20% at highest risk as predicted by the protein model had lower survival rates than the 20 at highest risk predicted by the baseline model, and the 20% predicted at lowest risk by the protein model had higher survival than those predicted at lowest risk by the baseline model (Supplementary Fig. 8).

**Different causes of death.** A visual examination showed that all models were reasonably well-calibrated, allowing predicted risk values to be interpreted directly as probabilities (Supplementary Fig. 9).

We also examined the difference in the performance of the prediction models for different causes of death. Figure 3 shows the predicted 5-year risk of all-cause mortality split by survival status after 5 years from plasma collection. Participants who died within 5 years are also shown separately for five cause-of-death categories; neoplasms, nervous system, circulatory system, respiratory system, and other. All five cause-of-death categories

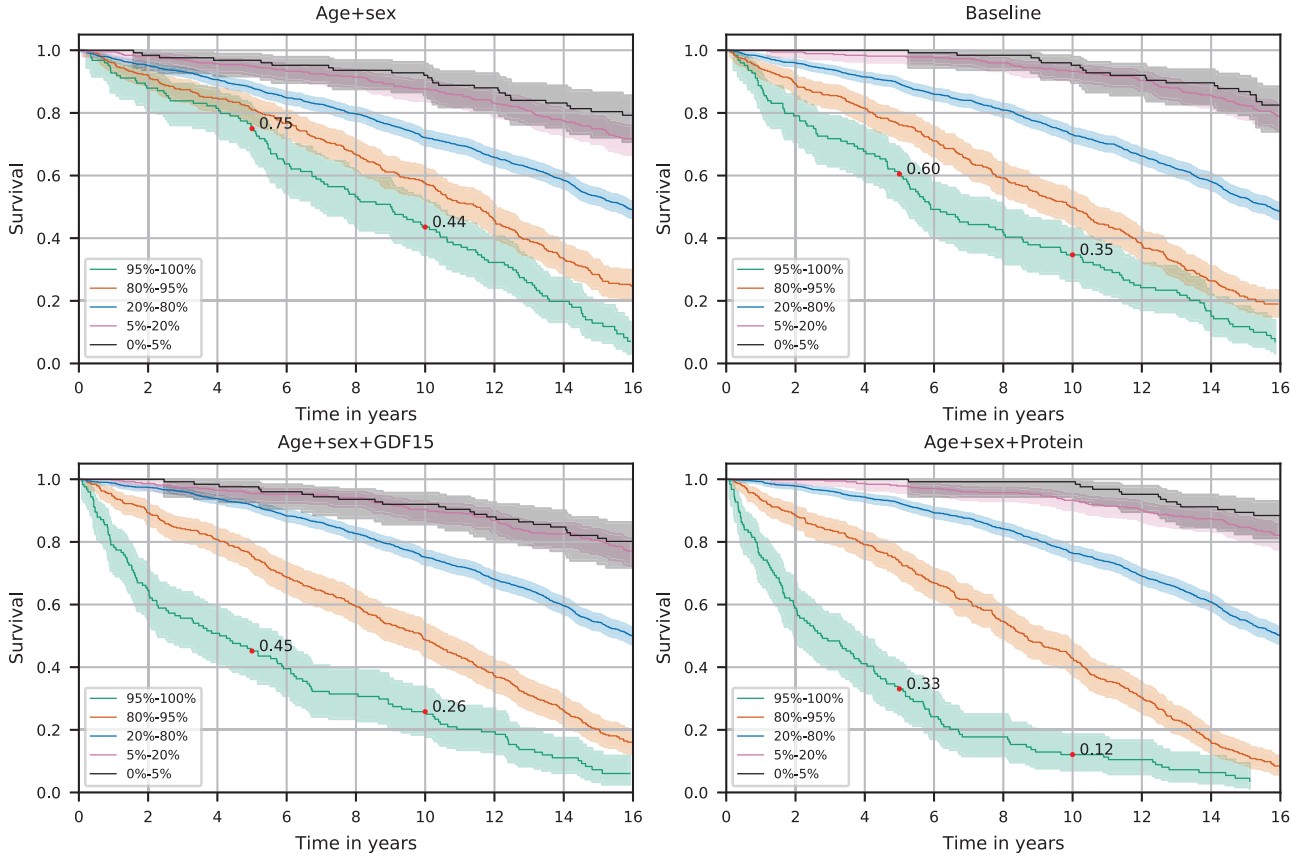

**Fig. 2 Survival of 60-80 years old participants.** The Kaplan–Meier curves for 2488 participants are split by quantiles of predicted 10-year risk by each model, demonstrating the different survival rates in the different risk groups. The colored areas represent 95% confidence intervals. The red dots show survival after 5 and 10 years.

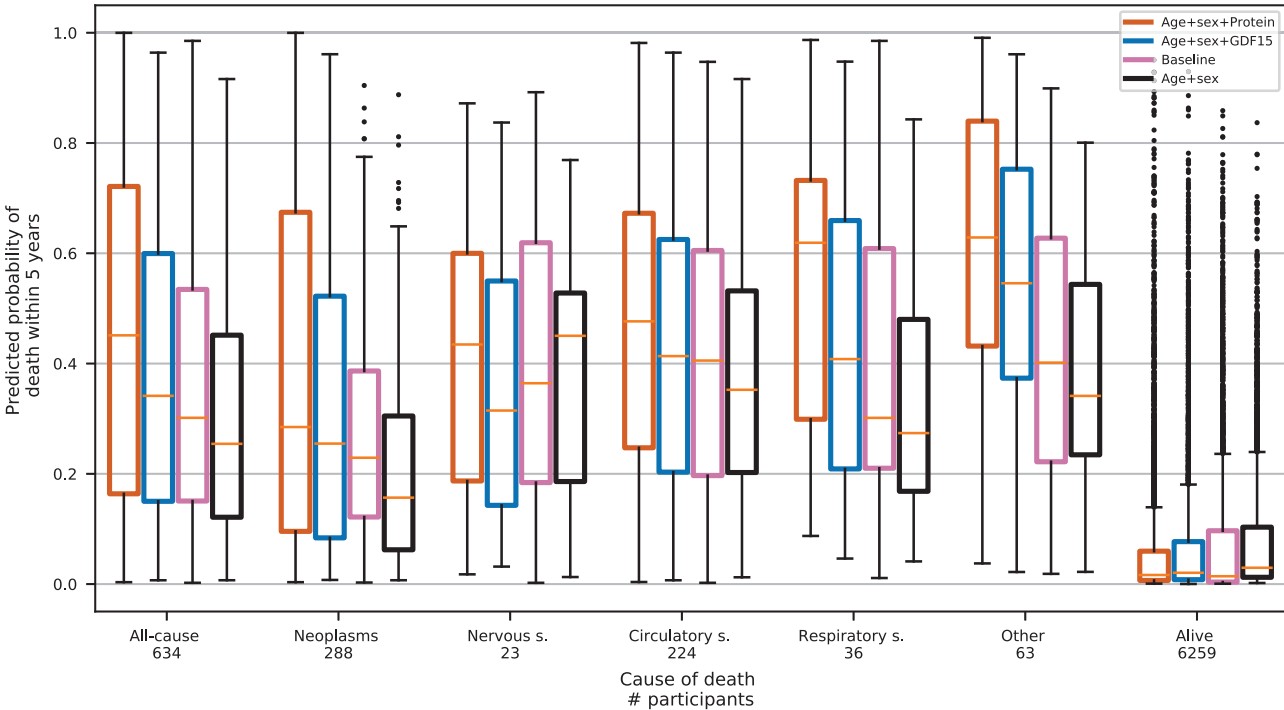

**Fig. 3 Predicted risk of death within 5 years for different causes of death.** Participants who died within 5 years are shown as one group and categorized by cause of death and those who were alive after 5 years are shown as one group. The yellow center line represents the median, the box limits are the upper and lower quartiles, the whiskers represent the 1.5× interquartile range, and the black dots are data points outside the whisker range.

showed a higher predicted risk for the protein model than the baseline model, although the difference varied between categories. Deaths from neoplasms were not as reliably predicted as deaths from other causes. Deaths from respiratory system causes and other causes showed the greatest improvement of the protein model over the baseline model. The predicted risk for those who did not die within 5 years was lowest with the protein model. We also looked directly at the AUC for each cause of death within 5 years, excluding other causes (Supplementary Fig. 10). There the protein model had the highest AUC for every category, best for the respiratory system and other causes, and worst for neoplasms.

**Associations of individual protein measurements**. In a univariate association corrected for age and sex, 1364 protein measurements out of 4905 were significantly associated with death within 5 years after a Bonferroni correction, 1292 positively and 72 negatively. For participants older than 60 years, the number of associated protein measurements was 1068, 1002 associated positively, and 66 negatively. Of the top ten single protein associations with death within 2, 5, 10, or 15 years, six proteins were common for all time points, with GDF15 always having the largest effect (Supplementary Table 4). As an example of protein measurement in our dataset we show the distribution and association of GDF15 with age in Supplementary Fig. 11. The distribution of GDF15 levels is close to log-normal and GDF15 levels seem to increase log-linearly with age. Most of the proteins with the highest associations were positively correlated. For the top ten proteins associating with all-cause mortality within 5 years, most intercorrelations after correcting for age and sex were 0.4–0.6 (Supplementary Fig. 12). A notable exception is ANTXR2, which was not correlated with any other top protein and the only top protein negatively associated with death. The average correlation between any protein measurement pair of the 4905 measurements after correcting for age and sex was 0.35.

When univariate associations were examined separately for different causes of death (Supplementary Table 5), the top two proteins for all-cause mortality, GDF15 and WFDC2, were among the top five proteins for all the death categories except for the nervous system category. The nervous system category had a different protein profile from the other categories, with the top four proteins having negative associations with death and GDF15 far from being significantly associated with death. This difference is probably partly responsible for the slight improvement of the protein model over the age and sex model in predicting nervous system deaths, as depicted in Fig. 3. All top five proteins for neoplasms were also in the top ten for all-cause mortality. Because neoplasms are the most common cause of death in our data, it is not surprising that all-cause mortality and neoplasm mortality have similar protein profiles.

Single protein predictor performance was in accordance with the univariate associations. Supplementary Fig. 5b shows that ANTXR2 was the top protein in predicting nervous system-related deaths. WFDC2 was best at predicting respiratory system-related deaths, while GDF15 was best at predicting neoplasm and other deaths. GDF15 and WFDC2 were best at predicting circulatory system deaths.

All 5-year all-cause mortality protein associations are in Supplementary Data 2 and Supplementary Data 3. We also examined associations using the Cox proportional hazards model with similar results (Supplementary Data 4).

**Pathway analysis**. We performed pathway analysis, finding the most over-or underrepresented protein pathways from the Reactome database[27], in the subset of most relevant proteins, selected with the Boruta method, for 5-year mortality prediction.

The subset includes 249 protein measurements (244 different proteins are included in the analysis) relevant for 5-year mortality prediction without correcting for age and sex. Using a false discovery rate of 0.1 as a significance threshold, three pathways were significantly overrepresented; Regulation of Insulin-like Growth factor (IGF) transfer and uptake by Insulin-like Growth Factor Binding Proteins (IGFBPs) (R-HSA-381426), Extracellular matrix organization (R-HSA-1474244), and Degradation of the extracellular matrix (R-HSA-1474228) which is a component of the Extracellular matrix organization pathway.

We repeated the pathway analysis in a set of most relevant protein measurements corrected for age and sex before applying Boruta. In the corrected subset, we have 246 protein measurements (238 different proteins are included in the analysis), of which 135 intersect with the subset from the uncorrected analysis. Using the same significance criteria, no pathway was significantly over-or underrepresented. However, the top overrepresented pathway was the Regulation of IGF transfer and uptake by IGFBPs ($p = 5.3e-5$). Other pathways with low p-values included many components of the Extracellular matrix organization pathway.

When we used the full set of 1364 protein measurements that were significantly associated with 5-year mortality (1314 different proteins are included in the analysis), no pathway was significantly over-or underrepresented, but Regulation of IGF transfer and uptake by IGFBPs and Extracellular matrix organization were among the three with the lowest $p$-values.

The ten pathways with the lowest $p$-values for all sets are in Supplementary Data 5.

**Ranking of included protein measurements**. Few protein measurements in addition to age and sex can achieve most of the discrimination performance. Five, ten, or twenty proteins, selected with a forward selection, yielded AUCs of 0.905, 0.910, and 0.912 for 5-year prediction and 0.913, 0.917, and 0.919 for 10-year prediction (Supplementary Fig. 13). In both 5- and 10-year prediction, the first three proteins selected into the model were GDF15, ANTXR2, and IGFBP2 (Supplementary Table 6).

**Heritability**. There were 20,983 sibling pairs and 18,166 parent-offspring pairs in the combined ICP, VSP1, dHS, and VSP2 datasets. Sibling pairs older than 60 years were 7022, and parent-offspring pairs in this age group were 1712. The sibling estimated heritability for predicted all-cause mortality risk was 0.22 and 0.24 in each age group and 0.13 and 0.17 when estimated with parent-offspring pairs. This method fails to account for similarities in the environment of relatives, making this an upper bound of heritability.

**Associated phenotypes**. The 8814 participants from the dHS underwent deep phenotyping at the time of sample collection. Although only 0.3% of them have died ($n = 25$) since recruitment, we calculated for all 8814 their predicted 5-year risk of all-cause mortality using the protein all-cause mortality prediction model, corrected it for age and sex, and correlated that with ten health and frailty related phenotypes. The predicted risk correlated negatively with the maximum oxygen uptake (VO2 max) in a graded cycle ergometer exercise tolerance test, max grip strength, forced expiration volume in one second (FEV1), number of correct codes in a digit coding test[28], and lean appendicular body mass scaled to height squared measured with Dual-energy X-ray absorptiometry (DXA). The predicted risk correlated positively with time spent completing trail making test B[29], resting heart rate, and average length from neck to waist over the

**Table 2 Correlation of 5-year mortality risk predicted by the protein model and corrected for age and sex with frailty related phenotypes in the dHS dataset.**

| Phenotype | All participants | | | Participants older than 60 | | |
|---|---|---|---|---|---|---|
| | N | Correlation | P-value | N | Correlation | P-value |
| Graded cycle ergometer exercise test: VO2 max | 6930 | −0.15 | 3.5E−36 | 2334 | −0.23 | 3.4E−28 |
| Max grip strength corrected for height | 8737 | −0.10 | 1.1E−21 | 3637 | −0.16 | 1.5E−21 |
| FEV1 | 8015 | −0.15 | 1.3E−40 | 3217 | −0.18 | 1.5E−25 |
| Digit coding: number of correct codes | 8562 | −0.09 | 6.1E−17 | 3530 | −0.14 | 1.5E−17 |
| Trail making test B: time to complete | 8485 | 0.09 | 2.7E−16 | 3475 | 0.10 | 1.1E−09 |
| Resting heart rate | 6688 | 0.08 | 2.0E−10 | 2851 | 0.08 | 6.0E−05 |
| Average length from neck to waist over back adjusted for height | 8022 | 0.08 | 7.0E−12 | 3300 | 0.12 | 1.0E−11 |
| Lean appendicular body mass divided by height squared | 8711 | −0.11 | 2.8E−23 | 3643 | −0.16 | 5.9E−22 |
| Non-HDL cholesterol level, not using statins | 6397 | −0.01 | 6.2E−01 | 2181 | 0.00 | 9.0E−01 |
| Non-HDL cholesterol level, using statins | 1525 | −0.03 | 1.9E−01 | 1181 | −0.04 | 1.7E−01 |
| BMI | 8812 | 0.03 | 8.7E−03 | 3683 | 0.01 | 5.5E−01 |

back. BMI and non-HDL cholesterol levels were not found to be correlated with predicted mortality risk (Table 2).

Information on various diseases and other traits collected through the Icelandic health system was available for most participants. We looked at how six traits, known to be risk factors for mortality, were associated with the predicted 5-year risk corrected for age and sex in the combined dHS and VSP2 datasets. The protein model predicted higher mortality risk for participants with type 2 diabetes (T2D), MI, CAD, or cancer, and those who smoked, but predicted risk did not associate with clonal haematopoiesis (Supplementary Table 7). For those who died during the study period ($n = 108$), the protein model predicted a much higher 5-year risk than the baseline model (Supplementary Fig. 14). The baseline had a C-index of 0.900, while the protein model had 0.938. In this case, we use C-index rather than AUC because of extensive censoring from limited follow-up in this dataset.

## Discussion

By analyzing 4905 measurements of 4684 plasma proteins in 22,913 participants, we developed a predictor, built on age, sex, and 81 to 219 protein measurements, that outperforms a predictor composed of traditional risk factors both for long- and short-term prediction. This was true both for participants 18 years or older and when restricted to participants over 60 years of age.

Adding the plasma protein GDF15 to the age and sex model yielded a model superior to the baseline model that included traditional risk factors. No other single protein surpassed the baseline. This supports previous research where GDF15 has been identified as an important biomarker of all-cause mortality[12,18,30]. It is also strongly associated with ageing[17,19,20], and associations have been found with cardiovascular diseases, cancer, diabetes, fibrosis, body weight, energy balance, and inflammation[12,31,32]. Although GDF15 is strongly associated with ageing, various diseases, and mortality, both positive and negative effects of high GDF15 expression have been reported[31]. It has been suggested that high GDF15 levels are protective responses against ageing and stress[31,33]. Further research is needed to determine the function of GDF15 in ageing and mortality risk to determine possible uses of it as a drug or drug target. Still, a high expression seems indicative of poor health.

We also showed that better performance could be achieved by adding more proteins to the age and sex model. The proteins were selected by the model without any consideration of their biology. The best short-term prediction model had fewer proteins than the best long-term prediction model. This may be partly explained by

the availability of more cases for longer-term prediction, making it possible to utilize more features without overfitting. Our long-term prediction also includes short-term prediction; therefore, both long-term and short-term risk factors have to be considered, making it a more complicated problem.

In a group of 60–80 years old, the protein model could identify a group of 5% with an 88% probability of dying within 10 years and a 67% probability of dying within 5 years. Furthermore, the protein model could identify a 5% group with a 1% probability of death within 10 years. In contrast, a similar high-risk group identified by the baseline model had a 65% probability of dying within 10 years and a 40% probability of dying within 5 years, while a 5% low-risk group had a 5% probability of dying within 10 years. This shows that with the protein model, a group at extremely high risk of death and another at very small risk can be identified. The protein model also relied much less on age in separating high- and low-risk groups than the baseline model. The difference in age between the groups was less for the protein model than the baseline model and the variance in age in each group higher.

The protein model predicted mortality within 5 years more accurately than the baseline model for various causes of death, i.e., neoplasms, the nervous system, the circulatory system, the respiratory system, and other reasons. Similarly, the model predicted those who did not die within 5 years at a lower risk. This suggests that the protein model predicts all-cause mortality rather than being biased to a specific cause of death. Of the various causes of death, the poorest prediction by all models was for cancer death. Since the age and sex model did considerably worse in that category, we suspect the poor performance is largely because the average age of death from neoplasms is lower than from any of the other causes.

It is reassuring to see that predicted risk by the protein model correlates well, in an independent dataset, with phenotypes that can be considered as measures of health and frailty[34]. Participants at higher predicted mortality risk performed worse in an exercise test, had weaker grip, lower FEV1, performed worse on a digit coding test, took longer time at a trail making test, had faster-resting heart rate, and had less appendicular lean body mass. Interestingly, the higher predicted risk was also correlated with greater length from neck to waist over the back, which could be due to age-related kyphosis[35].

We identified more than a thousand proteins associated with all-cause mortality, confirming many that have been identified before[12,13,18] and identifying new ones. It is interesting to note that about 5% of the associated proteins were negatively associated with death. Previous studies have shown that most plasma

protein levels rise with age, especially after 60[20]. We speculate that these associations have a common root and are likely to be either caused by the body starting to fail or as a response to counteract the failing.

Examining protein profiles of different disease categories of death revealed that most major systems, except the nervous system, have similar profiles. It is fascinating to find that GDF15, which has the biggest effect by a large margin in most categories, is not associated with nervous system deaths. Despite these differences, the top associations with nervous system deaths are all associated with all-cause mortality and in the same directions. From this, it seems that deaths connected to the nervous system lack factors common to other systems.

Pathway analysis on a set of proteins relevant for all-cause mortality prediction without correcting for age and sex points to involvement of the Regulation of IGF transfer and uptake by IGFBPs and the Extracellular matrix organization pathways. When the selection of relevant proteins was corrected for age and sex, no pathway was significantly overrepresented. However, the same pathways as for the uncorrected protein selection were among the most overrepresented. It is probable, since age is an important predictor of all-cause mortality, that these pathways are mostly connected to normal ageing. However, since they were also among the most important pathways when we corrected for age they might also be involved directly in mortality risk. Accelerated ageing, where biological age is higher than chronological age, is associated with increased mortality risk[18,19] and it seems likely that the same pathways are involved in normal ageing and accelerated ageing. The extracellular matrix provides structural support for the organs and is also involved in many other functions within the body[36]. The Extracellular matrix organization pathway has been connected with ageing and all-cause mortality before[18,20]. IGFs are major growth factors responsible for stimulating the growth of all cell types and are required for normal growth and health maintenance. IGFBPs are the main IGF transport proteins in the bloodstream, where they carry the growth factors predominantly in stable complexes[37].

One of the study's limitations is that some of the risk factors were not available at the time of plasma collection for some participants. Therefore some imputation had to be employed for medication data, BMI, and smoking. Furthermore, we used ApoB levels as a surrogate for non-HDL cholesterol and hypertension medication for blood pressure. Other common risk factors such as levels of creatinine, glucose, and triglycerides were not included in the baseline model, but our analysis suggests they would not have added much to the baseline. The baseline model would have benefited from the inclusion of other diseases than T2D, CAD, MI, stroke, and cancer. The severity of diseases, diagnosis times, and all medication information would probably also have improved the baseline model.

Another possible problem is that the training and testing data are enriched with cancer patients. Therefore, it is not a random sample of the population. However, the model could predict other causes of death better than that of cancer.

The model using age, sex, and protein levels outperformed the baseline model without having direct information about traditional risk factors. Thus, the protein approach only needs single blood draw to get prediction accuracy better than a model that includes multiple risk factor measurements and disease diagnosis. Recent technical advantages in simultaneously measuring a large number of proteins open up the possibility of accurate evaluation of an individual's state of health from only one blood draw. If the number of proteins is a limitation, only measuring 1-20 proteins still yields a powerful predictor.

The largest multi-biomarker all-cause mortality study we know of is the metabolomics study by Deelen et al.[8] They use a dataset of 44,168 individuals, which is far greater than ours, but in our dataset, we have more deaths during the study period and a larger set of biomarkers measured. We have not seen proteomics all-cause mortality studies with as many participants as we have. Therefore, we believe our study is the largest to date.

Other studies have identified combinations of proteins predictive of mortality in much smaller datasets[12,18,19]. These we could replicate to some extent in our data, confirming the predictive power of these protein combinations. Our protein predictor outperformed the others, mostly because it uses a much larger number of proteins. Our predictor also had the advantage of being developed in data from the same population as the test set.

A good all-cause mortality predictor could be useful to help assess treatment effects. It could, for example, be used as a clinical study endpoint, making it possible to get results without waiting for participants to die. Since our study's protein measurements are not normalized on the sample set, our predictor could be used directly on individual protein measurements from the SOMAscan platform. This was not possible with the metabolomics all-cause mortality predictor[8].

Death is the final event and can never be considered trivial. Any further insights into the long-term causes of death will always be valuable. This study shows the power of protein levels in plasma as predictors of death. Possible next steps could be to analyze the plasma proteins in terms of specific causes of death, finding causal relationships, and useful biomarkers for early detection of different health problems and the possibility of intervention.

## Methods

**Study populations**. The participants were all Icelandic, and the plasma samples were collected at two time periods. In the first dataset, 22,913 participants were recruited in the years 2000–2006 at deCODE through the Icelandic cancer project (ICP)[23] ($N = 20,226$) and various smaller projects (VSP1) ($N = 2687$) (Supplementary Fig. 1). The second dataset consists of participants recruited through the deCODE health study (dHS)[24] ($N = 8814$) in the years 2016–2019, and participants recruited in various smaller projects (VSP2) ($N = 6798$) at deCODE in the years 2010–2019. Since very little follow up (mean 1.4 years, sd. 1.1 years) was available for the 2010–2019 samples, we only used the samples from 2000–2006 (mean follow up 13.7 years, sd. 4.7 years) for the development of models to predict both long-term and short-term all-cause mortality.

Deaths and the causes of death until the end of 2018 were obtained from the Icelandic death registry. Cancer information was obtained from the Icelandic cancer registry, and previous cancer diagnosis was defined as any cancer diagnosis except non-melanoma skin cancer, which was excluded. Information on the diagnosis of T2D, CAD, MI, and stroke was gathered from the Icelandic healthcare system as has been described previously[38]. The Icelandic prescription registry provided medication information.

Pregnant women ($N = 145$), individuals who died from external causes (ICD10 S00-T98, $N = 241$), and participants younger than 18 were excluded from the study.

The cause of death was documented with ICD-10 codes. Codes C00-D48 are connected to neoplasms, G00-G99 to the nervous system, I00-I99 to the circulatory system, J00-J99 to the respiratory system, and all other codes are taken as one category of other causes.

All participants who donated samples gave informed consent, and the National Bioethics Committee of Iceland approved the study, which was conducted in agreement with conditions issued by the Data Protection Authority of Iceland (VSN_14-015, VSN_15-130, and VSN_15-214). Personal identities of the participant's data and biological samples were encrypted by a third-party system (Identity Protection System), approved and monitored by the Data Protection Authority.

**Protein measurements**. Blood was collected in EDTA tubes. The tubes were inverted 4–5 times before being centrifuged for 10 min at 3000$g$ at 4 °C. Plasma samples were frozen in aliquots at −80 °C. Plasma aliquots were kept away from light while they were allowed to thaw on ice. The aliquots were mixed by inverting the tubes three times and then centrifuged for 10 min at 3220$g$ at 4 °C before measurement.

All samples were measured with the SOMAscan platform (https://www.somalogic.com/), containing 5284 aptamers providing measurements of the relative binding of the plasma sample to each of the aptamers in relative fluorescence units (RFU). The technology and its performance have been previously described[14,39–42].

As a quality control, we calculated the correlation of log-transformed RFU units over all the 5284 aptamers for every pair of samples. We then calculated the average correlation of each sample with all other samples. The average correlation was high (median = 0.94), and we excluded samples with a correlation of less than 0.82. Furthermore, for evaluating the internal repeatability of the SOMAscan platform, we examined 200 samples drawn from the same individuals at different time points and 228 that were replicates of the same sample. We used the replicates to exclude aptamers that were not robust within the same sample. We also excluded aptamers that did not measure human proteins resulting in a total of 4905 protein aptamers measuring 4684 different proteins, i.e. unique UniProt IDs. Some aptamers measured multiple proteins, and some proteins were measured by multiple aptamers. In our dataset, to maintain consistency, we restricted the data to one sample per person. In the case of repeated measurements on the same individual, we chose the newest sample, and in the case of replicated measurements of the same sample, we selected one at random.

All protein levels were log-transformed. We randomly split the ICP and VSP1 data 70% / 30% into training/test sets. The means and standard deviations of the training set were used to standardize all features used for prediction. Only the training set was used for feature and model selection.

**Features of the development set**. If possible, all features were recorded/collected at the time of plasma collection. We included age, age squared, sex, and their interactions in all models. The baseline model also had current smoking status, T2D, CAD, history of MI, history of stroke, previous cancer diagnosis, use of statins, hypertension treatment, BMI, BMI squared, ApoB, and ApoB x statin. Most of the BMI values were available at the time of plasma collection, but 2095 were imputed with the median of all available BMI measurements for that individual, and BMI values for 455 individuals with no BMI measurements were imputed with the mean value of training and testing data separately. Since cholesterol levels were only available for a small portion of the participants, the ApoB protein was used as a substitute[43]. The ApoB protein has a correlation of 0.7 with non-HDL cholesterol in the dHS data, where both are available. Medication information was only available from 2003 and on, therefore statin use before 2003 was predicted using the proteomics data. Those receiving hypertension treatment in the first half of 2003 with samples collected earlier were assumed to have already been receiving treatment at the time of sample collection. Current smoking status was estimated using the proteomics data when data were not available. See Supplementary Data 6 for a summary of available measurements and performed imputations.

Additionally, baseline variable interactions with age and sex that had a p-value lower than 0.1 in logistic regression models for death within 1, 5, 10, or 15 years were included in the baseline model. Those were BMI, BMI squared, CAD, MI, cancer, statin use, and hypertension treatment interactions with age and CAD, smoking, cancer, and diabetes interactions with sex.

Other disease diagnoses were not considered, and missing values prevented the use of other quantitative features. We did, however, examine the effects of adding other quantitative features to the baseline in subsets with available data (Supplementary Fig. 15 and Supplementary Fig. 16). Common risk factors such as cholesterol levels (HDL, TC), systolic blood pressure (SBP), creatinine levels, and glucose levels were not found to add much to the baseline. Some combinations of features were found to add considerably to the baseline. However, there is probably some selection bias in the data since these measurements came from hospital data and the features were probably only measured in people where the measurements were thought to be important.

Polygenic risk scores (PRSs) for cancer, hypertension, stroke, CAD, Alzheimer's, attention deficit hyperactivity disorder (ADHD), Parkinson's, educational attainment, depression, bipolar disorder, BMI, schizophrenia, IQ, autism, and anorexia were tested as features in the all-cause mortality risk predictor. They did not improve the baseline prediction for 5-year mortality and only improved the 10-year prediction slightly (Supplementary Fig. 17). Based on this, we think that including PRSs will have more value in longer-term predictions. Due to their small effect, the PRSs were not used in any of our prediction models.

The Boruta feature selection method[25] was used to select all the most relevant protein measurements out of the available 4,905. This was done separately for events within 1,2,...,15 years.

**Metrics**. For comparing prediction performance, we used the ROC curve, the area under the curve (AUC), and the integrated discrimination improvement (IDI)[44]. When the follow-up was heavily censored, we made use of the concordance index (C-index)[45] instead of the AUC. The confidence intervals for the AUC and IDI were obtained by bootstrapping with 1,000 iterations, and ROC curves were compared using the Delong method[46]. Kaplan–Meier curves with log-log confidence intervals were also examined. Calibration was assessed with a visual examination of actual incidence and predicted risk in 5% quantile groups. The test data were used for all model comparisons.

**Types of prediction models**. We evaluated six different protein prediction models for 5 and 10-year prediction; logistic regression with an L1 penalty[26], logistic regression with an L2 penalty, logistic regression with an elastic net penalty[47], multi-layered perceptron (MLP), XGBoost[48] decision trees, and a Cox survival

model with an elastic net penalty. We optimized the parameters for the MLP and XGBoost with a Bayesian optimization algorithm and 5-fold cross-validation (CV), while we used a grid search and 5-fold CV for the rest. The parameters were chosen to minimize log-loss except in the Cox models, where the concordance was maximized. The methods were all compared using the mean AUC of 10-fold CV on the training set. Logistic regression performed best, where the penalty type did not have much effect on the result. We used logistic regression with an L1 penalty for the final model since it used the fewest features (Supplementary Fig. 18). For the age and sex model, baseline model, and age, sex, and one protein model, logistic regression was used. L2 penalty was added when prediction with age, sex, and 1–100 preselected proteins was performed.

In addition to models trained on all participants, separate models were trained for participants older than 60 years only. All prediction analysis restricted to the older than 60 group used predictions by these models.

**Other protein models**. We trained new predictors using logistic regression with PAD or the selected sets of proteins with age, sex, age squared, and their interactions or the baseline as features. We also experimented with using Cox proportional hazards models since those were used in the original predictors, but the logistic regression approach gave better predictions.

To predict biological age using the proteome, we used a linear regression model with L1 penalty. All 4,905 protein measurements and sex were used as candidate features in the model. The penalization strength was selected to minimize the mean square error using 5-fold CV on the training data. The PAD was then calculated as the difference between predicted age and chronological age.

We used all seven proteins identified by Tanaka et al.[18] since they were all available in our data. Our data did not include measurements of AGP1 and UCMGP. Therefore, only ten of twelve proteins identifies by Ho et al.[12] were used. They also included multiple covariates, not all available in our data, which prompted us to add the baseline features to the model.

**Univariate associations**. Associations of single protein measurements with mortality were examined using logistic regression. The model included age, age squared, sex, and their interactions as covariates. Associations were also examined using Cox proportional hazards models. The Cox model used age, age squared, sex, and their interactions as covariates. To avoid age violating the proportional hazards assumption significantly, the model used different baseline hazards for each age bin; 18–40, 40–60, 60–80, and 80+. Associations were considered significant if they had a p-value lower than 0.05 after Bonferroni correction, i.e., lower than 0.05/4905 = 1.02e−5. Associations with cause-specific mortality were examined by excluding deaths from all other causes in the data. Correlations of protein measurements were calculated with Pearson correlation after correcting for age, age squared, sex, and their interactions.

**Pathway analysis**. For analyzing the over- and underrepresentation of Reactome[27] protein pathways, we used the PANTHER classification system, version 16 (http://pantherdb.org/)[49]. As a reference, we used all 4905 protein measurements, which resulted in 4619 unique Uniprot IDs recognized by the system. In cases where a protein measurement had multiple proteins associated with it, we included all of them. The Fisher exact test was used to determine statistical significance and false discovery rate (FDR) to account for multiple testing. To select proteins relevant to 5-year mortality risk the Boruta[25] feature selection method was used. To correct for age and sex we subtracted linear regression of age, age squared, sex and their interaction from the protein levels before applying Boruta.

**Ranking of protein measurements**. We used two approaches to order protein measurements by effects. The first method was a stepwise forward selection with age, age squared, sex, and their interactions as baseline features, sequentially adding the protein that maximally increased the log-likelihood. The other method trained 1000 logistic regression models with an L1 penalty each time resampling the training data. The protein measurements were then ordered by how often they were included in the model. In the case of protein measurements included equally often, the protein measurement with the higher mean coefficient was ranked higher.

**Heritability estimate**. Heritability of 5-year predicted mortality risk was estimated by using the correlation of mortality risk between siblings and between parent-offspring pairs. The predictions were corrected for age and sex, normalized, and corrected again for age, sex, and year of birth.

**Associations with phenotypes**. Before examining connections to other phenotypes, the predicted values were corrected for age, age squared, sex, and their interactions using linear regression. Calculated correlations are Pearson correlations, and means were compared with a two-sided t-test. Associations were considered significant if they had a p-value lower than 0.05 after Bonferroni correction. The quantitative phenotypes were corrected for age and sex and normalized.

**Statistics and reproducibility**. Standard metrics and statistical tests were used to evaluate and compare models. All data preparations, plotting, model training, and

most of statistical tests were done in Python version 3.6.3. The logistic regression models were implemented using the machine learning library scikit-learn[50]. AUCs were compared using R version 3.6.0 with the package pROC[51].

**Reporting summary**. Further information on research design is available in the Nature Research Reporting Summary linked to this article.

## Data availability

We declare that the data supporting the findings of this study are available within the article, its supplementary information, and upon reasonable request. Individual-level data used in this publication are not publicly available because Icelandic law and the regulations of the Icelandic Data Protection Authority prohibit the release of individual-level and personally identifying data.

## Code availability

Custom code is available at https://github.com/thjodbjorge/Mortality_prediction.

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

## Author contributions

T.E., D.F.G., P.S., U.T., K.S., and M.O.U. designed the experiments, interpreted the results and drafted the manuscript. T.E., S.A., B.A.J., S.H.L., E.V.I., K.N., E.F., D.F.G., and M.O.U. analyzed the data. H.S., I.J., H.H., T.R., J.S., G.L.N., G.T., U.T., and K.S. contributed to acquisition of the data and T.E., S.A., B.A.J., E.F., H.S., H.H., T.R., D.F.G., P.S., U.T., K.S., and M.O.U. revised the manuscript. All authors contributed to the final version of the paper.

## Competing interests

The authors declare the following competing interests: all authors are employed by deCODE genetics/Amgen, Inc.
