## [Peer Review File · Communications Biology]

Reviewers' Comments:

Reviewer #1:

Remarks to the Author:

Eiriksdottir et al, through comprehensive analysis using SOMAscan assay (~4,700 aptamers)' in the largest study so far (n=22,913), tried to predict the probability of death. 7061 deaths were reported in 13.7 (SD 4.7) years of follow up. The study successfully compared prediction of deaths using four main models (age and sex model, baseline model (age+sex+lifestyle/diseases), GDF15model (age+sex+GDF15) and protein model (age+sex+ proteins). Study successfully delineated participants group with higher and low risk of death. The longitudinal study design as well as larger sample size captured more deaths adding power to the study (almost third of participants succumbed to death in course of study). Predicted risk of death were correlated with frailty variables in an independent cohort.

Comments

1. Using proteomic of ~5000 proteins as predication for mortality in >22,0000 this paper does not make any effort to investigate the biology of mortality/aging. This paper does show top 10 proteins as related to age related disease and frailty. However, a pathway analysis would reveal which groups of proteins are over or underrepresented with aging. This is a great limitation of this paper that deals with prediction without biological content. In the last year, papers by Ferucci, Barzilai, and Tony Wyss-Corray have shed light and created better prediction models, in less subjects. These should also be mentioned as these data mainly validates them.

i. <https://elifesciences.org/articles/61073>

ii. <https://onlinelibrary.wiley.com/doi/full/10.1111/accel.13250>

2. This lack of biological connection is striking with the example of GDF 15. Overexpression of this peptide in animals lead to extended healthspan and life span and its levels in crease when an a gerotherapeutic like metformin is used in humans. So, this is most probably a protective protein and high levels are efforts to protect against aging, its important to note this content in the paper.

3. Authors try to convey that GDF-15 as well as protein models outperformed baseline model which is mainly based on life style and disease. Do authors think baseline model could be further improved by including other important features (triglycerides, glucose/HbA1c, blood pressure meares) as well as other measures?

4. Changes in AUC between groups is 0.0x, is that clinically significant? For example, GDF15 model for all-cause mortality had an AUC of 0.893 compared to baseline model (AUC=0.885) and "age and sex model "model (AUC=0.852), difference of . 0.030.

5. Can you present a sensitivity analysis excluding all main diseases at the baseline in predicting death? What about using a model excluding all comorbidities at baseline and predicting death?

6. What was the reason for using 60 years as cutoff for defining group as well as using 60-80 years for KM analysis? Does this age cutoff capture any specific aspect of aging in this population

7. AUC and ROC values decreases when "older than 60 years" only were analyzed compared to complete age range. There are more deaths reported in age group of 60 years and above (From table 1, there were 6222 deaths compared to only 839 deaths in 18-60 range). If there were more deaths happening in the age range, why wasn't it helping in modelling prediction. Some of the proteome is taking off after that age, does it influence the results?

8. What is the distribution of GDF15 across age and is there a drastic change in expression happening after 60 years (cutoff used in manuscript) where maximum deaths are reported? Also a plot showing distribution of GDF15 will be helpful.

9. From figure 3 we could conclude that inclusion of diseases and lifestyle (baseline) +proteins provide better strategy for predicting death. How important is a compilation of different features for prediction rather than separating them out.

10. All the recent proteomic studies have shown GDF15 to be one of the top hit proteins to be associated with aging. GDF15 is capturing the underlying biology is well supported where GDF15+Age+Sex out performs baseline model (Fig 3). Is there a possibility of GDF-15 capturing lifestyle associated biological changes better than actual phenotypic data on lifestyle? Though authors have used GDF15 as model across manuscript, what is the second-best alternative in capturing mortality?

11. What would a sensitivity analysis excluding GDF15 from all the protein model show?
12. Using cox survival analysis as sensitivity analysis instead of logistic regression for univariate associations could be helpful. Time to death could be used instead of 5 years cutoff.
13. It is interesting to note that top single protein association with Nervous system (Supplementary Table 6) are having negative Beta, while most of other systems top proteins have positive Beta value. What are the possible explanations? Also, what are the possible explanations for little success in capturing nervous system associated death. Disease progression time as well as severity of the diseases varies. How much does it influence the prediction modeling?
14. What are the possible explanations of PRS not having a drastic effect in prediction modeling?

Minor comments

1. Why was L1 penalty selection of 219 proteins for 13 years mentioned in line 87 than 192 for 15 years?
2. Line 112 mentions supplementary Figure 1. Instead of supplementary Figure 2. Supplementary figures number should be redone as there are changes at multiple places.
3. Line 167 "to and age and sex" please check
4. Lines 248-249 please rephrase the sentences

Extensive proteomic data used in this study will definitely provide important contribution in understanding basis of complex phenotypes of aging as well as age associated diseases.

Reviewer #2:

Remarks to the Author:

The authors set out to predict mortality based on levels of thousands of proteins in plasma from more than 20,000 individuals age 18-101. They develop various models using demographic and other lifestyle assessments in combination with proteomics to develop risk assessments for various times into the future. Overall, the size of the cohort and the number of proteins measured are most impressive and an incredibly valuable treasure trove and the reported results are clear.

What this reviewer missed was an interpretation of the findings, biological significance of the observed top predictor proteins, a discussion of the potential utility of the findings, and a comparison with all the other mortality predictors out there. After reading the article I was not clear what the next steps were and why the authors carried out the study. The final statement in the abstract " Our results show that the plasma proteome can be used to estimate general health and the risk of death" is rather non-specific and, as the authors discuss in the introduction, not novel.

Major questions:

- 1) The study combines multiple cohorts and it appears they have not been merged or normalized. In similar studies using the Somalogic platform, a simple PCA revealed the origin of samples based on cohort or center of collection before normalization (some studies did not normalize and analyzing their data reveals this major flaw). This needs to be addressed. It is not stated how samples were collected (EDTA, heparin, citrate) but I assume they were of the same type?
- 2) Related to 1) can the authors build a model in one cohort and predict mortality with similar accuracy and using the same predictors in an independent cohort?
- 3) I had a hard time reading this article as it basically recites all the numbers and p-values which are, or which should be, in figures - hence we use graphics for scientific presentations. It would be more important to tell the reader what the results mean: why are these specific proteins selected into the predictor, why are there different ones for the various diseases? are they in specific pathways or does it mean anything biologically? have other studies identified similar biology as discussed in the introduction, what are the overlapping, consistent proteins, etc.?

We thank the reviewers and the editor for their comments and for taking the time to thoroughly read our manuscript. Below we have responded point-by-point to each comment. In doing so, we think our manuscript has improved considerably, and we hope that the reviewers agree. In addition to responding to the comments we have fixed some minor errors that we came across, changed the line colours of the figures, and added authors that helped with the revision or should have been included from the beginning.

The reviewers' comments are included as they were written but in bold letters, our answers are in italics, text from the manuscript is in italics and underlined. Reference numbers in all text from the manuscript relate to the manuscript references and new or modified text is coloured red. All major changes are coloured red in the revised manuscript.

Reviewer #1 (Remarks to the Author):

Eiriksdottir et al, through comprehensive analysis using SOMAscan assay (~4,700 aptamers)' in the largest study so far (n=22,913), tried to predict the probability of death. 7061 deaths were reported in 13.7 (SD 4.7) years of follow up. The study successfully compared prediction of deaths using four main models (age and sex model, baseline model (age+sex+lifestyle/diseases), GDF15model (age+sex+GDF15) and protein model (age+sex+ proteins). Study successfully delineated participants group with higher and low risk of death. The longitudinal study design as well as larger sample size captured more deaths adding power to the study (almost third of participants succumbed to death in course of study). Predicted risk of death were corelated with frailty variables in an independent cohort.

Comments

1. Using proteomic of ~5000 proteins as predication for mortality in >22,0000 this paper does not make any effort to investigate the biology of mortality/aging. This paper does show top 10 proteins as related to age related disease and frailty. However, a pathway analysis would reveal which groups of proteins are over or underrepresented with aging. This is a great limitation of this paper that deals with prediction without biological content. In the last year, papers by Ferucci, Barzilai, and Tony Wyss-Corray have shed light and created better prediction models, in less subjects. These should also be mentioned as these data mainly validates them.

i. <https://elifesciences.org/articles/61073>

ii. <https://onlinelibrary.wiley.com/doi/full/10.1111/accel.13250>

Thanks for pointing us to these papers. We agree that they are relevant to our paper and should be discussed in the manuscript. However, we do not agree that our study mainly validates these studies, though they certainly support each other. All three papers examine the associations between the plasma proteome measured using SOMAscan assays to mortality in some way. Both of the mentioned papers mostly focus on age-protein associations while our focus is on mortality prediction and mortality associations corrected for chronological age. The paper by Tanaka et al.¹ creates an all-cause mortality predictor with 7 proteins and compares the performance with single protein and age predictors. Sathyan et al.² only examine older adults and use predicted age to predict mortality. However, as mentioned by the reviewer, the longitudinal study design and our large dataset is probably our greatest contribution.

We have modified the Introduction to include the papers mentioned by the reviewer and three others³⁻⁵ that we came across recently.

Similarly, 56 peptides (31 proteins) correlated with five-year mortality in a study of 2,473 older men. A panel of those peptides improved the predictive value of a commonly used clinical predictor of mortality¹³.

With the advent of new technology such as SOMA¹⁴ or proximity extension assays¹⁵, it is possible to simultaneously measure levels of thousands of proteins efficiently. Several studies using this technology have shown the plasma proteome to be heavily associated with age and life span^{16–20}. A study of 997 participants associated 651 out of 1,301 proteins with age, found that a 76-protein proteomic age signature associated with all-cause mortality independent of chronological age, and created a seven-protein mortality predictor¹⁸. In a study of 1,025 older adults, 754 of 4,265 proteins were associated with age. A proteomic age model using the age-associated proteins predicted mortality better than chronological age¹⁹. Another study of 4,263 participants measured 2,925 proteins to evaluate how circulating protein profile changes over the lifespan²⁰. Some studies have used large proteomics datasets to predict other health-related factors. A protein-based risk score for cardiovascular outcomes in a high-risk group was developed using 1,130 candidate plasma proteins²¹. In addition, ~5,000 plasma proteins were used to predict health states, behaviour, and incident diseases, with performance comparable with traditional risk factors, in 16,894 participants²². These studies underscore the value of using plasma levels of a large number of proteins to search for biomarkers in health and diseases.

To integrate further previously published material in our manuscript we tried to replicate or use similar mortality prediction approaches as developed in these two papers and in that of Ho et al.⁶ and compared to our predictors. To this end we have added the following text to the Results section and Supplementary Figure 6.

Other protein predictors

Other protein-based mortality predictors have been developed. We tried to replicate them in our data and compared them to our predictors as shown in Supplementary Figure 6. The difference between protein derived age and chronological age, sometimes called predicted age difference (PAD), has been shown to be predictive of mortality^{18,19}. We calculated a PAD and used it as a feature in a mortality prediction model. The PAD was predictive of mortality but was far from being as good of a predictor as GDF15 alone. We also tried a mortality predictor using the seven proteins shown to be useful mortality predictors by Tanaka et al.¹⁸. The seven-protein model performed better than our GDF15 model but did not reach our protein model performance. This is expected since GDF15 was one of the seven proteins and because our protein model makes use of more proteins. Finally, we tried using ten of twelve proteins used in a multivariable model by Ho et al.¹², adjusted for many of our baseline variables. This model also included GDF15 and had prediction performance between our protein model and the seven protein model which might be expected since it used more proteins than seven and fewer than our protein model. These proteins were selected out of a set of proteins targeted because of their high value for CVD.

Supplementary Figure 6: AUC for protein models from other studies, at different time points, compared to our models. The figures compare our model which uses multiple proteins and different number of proteins for each time point, to models using combinations of proteins or protein derived PAD identified as good predictors in previous studies. a) Every model includes age and sex, b) every model includes the baseline.

In relation to this we have added the following text to the Discussion:

Other studies have identified combinations of proteins predictive of mortality in much smaller datasets^{12,18,19}. These we could replicate to some extent in our data, confirming the predictive power of these protein combinations. Our protein predictor outperformed the others, mostly because it uses a much larger number of proteins. Our predictor also had the advantage of being developed in data from the same population as the test set.

We have also added a better description of how these other predictors were implemented to the Methods.

Any attempt to examine the biology is indeed missing from this paper. We were more focused on the use of proteins as biomarkers for prediction rather than their biological effect. However, the reviewer's suggestion is valid and we have therefore included in the revised manuscript over/under-representation pathway analysis of proteins useful for mortality prediction. The following section has been added to the Results:

Pathway analysis

We performed pathway analysis, finding the most over-or underrepresented protein pathways from the Reactome database²⁶, in the subset of most relevant proteins, selected with the Boruta method, for five-year mortality prediction. The subset includes 249 protein measurements (244 different proteins are included in the analysis) relevant for five-year mortality prediction without correcting for age and sex. Using a false discovery rate of 0.1 as a significance threshold, three pathways were significantly overrepresented; Regulation of Insulin-like Growth factor (IGF) transfer and uptake by Insulin-like Growth Factor Binding Proteins (IGFBPs) (R-HSA-381426), Extracellular matrix organization (R-HSA-1474244), and Degradation of the extracellular matrix (R-HSA-1474228) which is a component of the Extracellular matrix organization pathway.

We repeated the pathway analysis in a set of most relevant protein measurements corrected for age and sex before applying Boruta. In the corrected subset, we have 246 protein measurements (238 different proteins are included in the analysis), of which 135 intersect with the subset from the uncorrected analysis. Using the same significance criteria, no pathway was significantly over-or underrepresented. However, the top overrepresented pathway was the Regulation of IGF transfer and uptake by IGFBPs ($p = 5.3e-5$). Other pathways with low p -values included many components of the Extracellular matrix organization pathway.

When we used the full set of 1,364 protein measurements that were significantly associated with five-year mortality (1314 different proteins are included in the analysis), no pathway was significantly over- or underrepresented, but Regulation of IGF transfer and uptake by IGFBPs and Extracellular matrix organization were among the three with the lowest p-values. The ten pathways with the lowest p-values for all sets are in Supplementary Table 12.

In relation to the pathway analysis we have added the following paragraph to the Discussion:

Pathway analysis on a set of proteins relevant for all-cause mortality prediction without correcting for age and sex points to significant involvement of the Regulation of IGF transfer and uptake by IGFBPs and the Extracellular matrix organization pathways. When the selection of relevant proteins was corrected for age and sex, no pathway was significantly overrepresented. However, the same pathways as for the uncorrected protein selection were among the most overrepresented. It is probable, since age is an important predictor of all-cause mortality, that these pathways are mostly connected to normal ageing. However, since they also came on top when we corrected for age they might also be involved directly in mortality risk. Accelerated ageing, where biological age is higher than chronological age, is associated with increased mortality risk^{18,19} and it seems likely that the same pathways are involved in normal ageing and accelerated ageing. The extracellular matrix provides structural support for the organs and is also involved in many other functions within the body³⁴. The Extracellular matrix organization pathway has been connected with ageing and all-cause mortality before^{18,20}. IGFs are major growth factors responsible for stimulating the growth of all cell types and are required for normal growth and health maintenance. IGFBPs are the main IGF transport proteins in the bloodstream, where they carry the growth factors predominantly in stable complexes³⁵.

A description of how this analysis was done was added into the methods section.

2. This lack of biological connection is striking with the example of GDF 15. Overexpression of this peptide in animals lead to extended healthspan and life span and its levels in crease when an a gerotherapeutic like metformin is used in humans. So, this is most probably a protective protein and high levels are efforts to protect against aging, its important to note this content in the paper.

In light of this comment, we have extended our sentence of previously identified associations of GDF15 with health and diseases to the following paragraph:

Adding the plasma protein GDF15 to the age and sex model yielded a model superior to the baseline model that included traditional risk factors. No other single protein surpassed the baseline. This supports previous research where GDF15 has been identified as an important biomarker of all-cause mortality^{12,18,28}. It is also strongly associated with ageing^{17,19,20}, and associations have been found with cardiovascular diseases, cancer, diabetes, fibrosis, body weight, energy balance, and inflammation^{12,29,30}. Although GDF15 is strongly associated with ageing, various diseases, and mortality, both positive and negative effects of high GDF15 expression have been reported.²⁹ It has been suggested that high GDF15 levels are protective responses against ageing and stress^{29,31}. Further research is needed to determine the function of GDF15 in ageing and mortality risk to determine possible uses of it as a drug or drug target. Still, a high expression seems indicative of poor health.

3. Authors try to convey that GDF-15 as well as protein models outperformed baseline model which is mainly based on life style and disease. Do authors think baseline model could be further

improved by including other important features (triglycerides, glucose/HbA1c, blood pressure measures) as well as other measures?

We think the baseline model could be improved by adding other features. The baseline could probably be made to surpass the protein model by adding enough features from different sources. We did, however, not have many features available for more than half the participants. We decided to add information on a previous stroke to the baseline and examine some available quantitative features further, focusing on the features suggested by the reviewer, as well as some used in other mortality prediction models. To this end we have added to the revised manuscript Supplementary Figures 15 and 16, Supplementary Figure 17 was changed, so it did not repeat information from the new figures. In the methods section some text on other baseline feature experiments was replaced with the following text:

Other disease diagnoses were not considered, and missing values prevented the use of other quantitative features. We did however, examine the effects of adding other quantitative features to the baseline in subsets with available data (Supplementary Figure 15 and Supplementary Figure 16). Common risk factors such as cholesterol levels (HDL, TC), systolic blood pressure (SBP), creatinine levels, and glucose levels were not found to add much to the baseline. Some combinations of features were found to add considerably to the baseline. However, there is probably some selection bias in the data since these measurements came from hospital data and the features were probably only measured in people where the measurements were thought to be important.

Supplementary Figure 15: Extra features in the baseline model. The blue dots show the increase of mean AUC from including the features on the x-axis in the baseline model. The bars show for how many participants the features are available. Since the AUC is heavily influenced by the dataset, the results for a protein model trained on the same data is also shown (red dots).

Supplementary Figure 16: Sets of extra features for the baseline model. The blue dots show the increase of mean AUC from including the set of features in the baseline model. The bars show for how many participants all the features in the set are available. Since the AUC is heavily influenced by the dataset, the results for a protein model trained on the same data is also shown (red dots). **Bloodcells**: Hematocrit, Hemoglobin, MCH, MCHC, Platelets, WBC, and RDW. **Likely3**: SBP, Glucose, and Triglycerides. **Top3**: ALP, ESR, and RDW. **High7**: ALP, ESR, RDW, Fibrosis-4, Hematocrit, Hemoglobin, and RBC. **Likely6**: SBP, Glucose, Triglycerides, HDL, TC, and Creatinine. **CV risk**: SBP, HDL, TC.

We also modified the discussion on the baseline factors in the Discussion:

One of the study's limitations is that some of the risk factors were not available at the time of plasma collection for some participants. Therefore some imputation had to be employed for medication data, BMI, and smoking. Furthermore, we used ApoB levels as a surrogate for non-HDL cholesterol and hypertension medication for blood pressure. Other common risk factors such as levels of creatinine, glucose, and triglycerides were not included in the baseline model, but our analysis suggests they would not have added much to the baseline. The baseline model would have benefited from the inclusion of other diseases than T2D, CAD, previous MI, previous stroke, and previous cancer diagnosis. The severity of diseases, diagnosis times, and all medication information would probably also have improved the baseline model.

4. Changes in AUC between groups is 0.0x, is that clinically significant? For example, GDF15 model for all-cause mortality had an AUC of 0.893 compared to baseline model (AUC=0.885) and "age and sex model" model (AUC=0.852), difference of 0.030.

Clinical significance would imply a very powerful tool. It is very hard, if not impossible, to estimate clinical significance at this stage. Therefore we don't make any claims in that direction. However, it is worth noting that though the difference between the models measured in AUC for the entire age range is not very big, it is perhaps not the best measure of clinical significance. In this case, we are comparing predictions for everyone from 18-101 where age is always going to order most of the people correctly in terms of mortality risk. It is more interesting to look at people that are more likely to be compared in a clinical setting such as the 60+ group where the difference between the age and sex model AUC and the protein model AUC was 0.103 and the baseline model and protein model 0.052. In the 60-80 year old group, the identified high-risk group by the protein model was 1.8 times more likely to die within five years than a similar group identified by the baseline model, which we think is a considerable difference. We also looked at the integrated discrimination improvement (IDI) which was 0.114 for the protein model over the baseline showing that the difference in the average of predicted risk between events and non-event was 0.114 higher for the protein model than the baseline model.

5. Can you present a sensitivity analysis excluding all main diseases at the baseline in predicting death? What about using a model excluding all comorbidities at baseline and predicting death?

We tested our model on a subset of the participants where MI, CAD, cancer, and previous stroke had been excluded. The results from this test were added as Supplementary Figure 3a, and the following text was added to the manuscript:

When the predictors were applied to a subset of participants not diagnosed with any of the major diseases used in the baseline at the time of plasma collection, the protein model was still the best prediction model (Supplementary Figure 3a). The baseline model and the GDF15 model still did better than the age and sex model, but the difference is much smaller than when the whole dataset is used. This is not surprising since information about the excluded diseases is essential to the baseline model.

Supplementary Figure 3: AUC at different time points in subsets of the test data. a) All participants not diagnosed with major diseases (Cancer, previous MI, previous stroke or CAD) at time of plasma collection.

We also tried excluding all smokers, medication users, and those with diabetes. This left 2961 participants in the test sets, of which 324 died during the study period. In this analysis, the baseline did not predict better than the age and sex model, the protein model was best and the GDF15 model was the second best. We did not include these analysis in the manuscript since we felt it did not add much.

6. What was the reason for using 60 years as cutoff for defining group as well as using 60-80 years for KM analysis? Does this age cutoff capture any specific aspect of aging in this population?

We used the 60-year cutoff because we had seen it used before⁷, and we thought it reasonable cutoff to exclude those in little danger of dying based only on age. According to Statistics Iceland, the yearly death rate in 60-65-year-olds in Iceland was 0.7% from 2000-2020 (<https://px.hagstofa.is:443/pxen/sq/23141d1c-071f-46dd-a407-1226dba8ccfb>). The cutoff at 80 was done because we wanted to examine a group with high variability in time until death. Participants over 80 almost all died within 15 years, making long term prediction in that age group less interesting. The exact cutoff ages were chosen rather arbitrarily at whole decades, aiming to separate old from middle age and very old from old.

7. AUC and ROC values decreases when "older than 60 years" only were analyzed compared to complete age range. There are more deaths reported in age group of 60 years and above (From table 1, there were 6222 deaths compared to only 839 deaths in 18-60 range). If there were more deaths happening in the age range, why wasn't it helping in modelling prediction. Some of the proteome is taking off after that age, does it influence the results?

We think the AUC and ROC values decreased when only 60+ were included because when we exclude participants younger than 60 we are mostly excluding participants that are easily classified. Most of the excluded participants are young and, therefore, healthier than the 60+ and don't die in the near future. Reduced power of the age and sex model in the 60+ group shows that age alone is not as powerful a predictor as with the whole age range. The difference between the protein model and the baseline model was bigger in the 60+ age group, probably because age is not as powerful a predictor in the smaller age range, and there are more diseases and frailty that can be detected with the proteome. A similar explanation was added to the manuscript:

The lower AUC and bigger AUC differences probably result from the exclusion of participants younger than 60 since they are easily classified. For example, the youngest age group is at very low mortality risk and thus easily distinguished using the age variable. That is, the smaller age range reduces the importance of age.

Any changes happening after 60 connected with normal aging should not influence the prediction more than age, which is included in the models. We speculate that some of the changes in the proteome after 60 are related to the body starting to fail and higher prevalence of many diseases with increasing age. These changes are exactly what we hope to catch with the protein model and likely to influence the prediction.

8. What is the distribution of GDF15 across age and is there a drastic change in expression happening after 60 years (cutoff used in manuscript) where maximum deaths are reported? Also a plot showing distribution of GDF15 will be helpful.

The relationship between the logarithm of GDF15 levels and age is close to a linear relationship and we did not observe drastic changes in the expression of GDF15 at any age. We added Supplementary Figure 14 to the manuscript showing the distribution of GDF15 in our data and the relationship between age and GDF15 levels. The following text has also been added to the Results:

As an example of protein measurement in our dataset we show the distribution and association of GDF15 with age in Supplementary Figure 11. The distribution of GDF15 levels is close to log-normal and GDF15 levels seem to increase log-linearly with age.

Supplementary Figure 11: GDF15 in the ICP+VSP1 dataset. a) Levels of GDF15 vs. age. The figure also shows the best straight line and a line fit with locally weighted scatterplot smoothing. b) The distribution of GDF levels.

9. From figure 3 we could conclude that inclusion of diseases and lifestyle (baseline) +proteins provide better strategy for predicting death. How important is a compilation of different features for prediction rather than separating them out.

We are not sure we understand this question correctly. Figure 3 is meant to show that our predictor is not only predicting death from one cause but predicts high probability of death from many different causes as well as predicting low probability of death for those who did not die. We added some text to the figure captions to make this clearer.

For predicting all-cause mortality a compilation of different features is important because all-cause mortality includes all-causes. A very good all-cause mortality predictors could probably be developed from medical records using a compilation of different clinical factors. This would however be very complicated since such data are complex and often incomplete. The protein predictor has the advantage of using only data from a single source, one plasma sample. For mortality prediction for a single cause fewer features should be necessary, making use of medical data easier.

We have seen from our protein association studies that most death causes have common factors, e.g. high levels of GDF15 and WFDC2. This is important for our all-cause mortality prediction and makes it easier than cause specific prediction, as there are more available mortality cases and a model trained to predict all-cause mortality does not need to learn to distinguish separate causes of mortality.

10. All the recent proteomic studies have shown GDF15 to be one of the top hit proteins to be associated with aging. GDF15 is capturing the underlying biology is well supported where GDF15+Age+Sex out performs baseline model (Fig 3). Is there a possibility of GDF-15 capturing lifestyle associated biological changes better than actual phenotypic data on lifestyle? Though authors have used GDF15 as model across manuscript, what is the second-best alternative in capturing mortality?

For some lifestyle variables, we think GDF15 might be capturing lifestyle associated biological changes better than phenotypic data. Phenotypic data are often inaccurate and difficult to collect. In some cases self-reporting by participants is the only way to evaluate the variables and quantitative variables are often treated as categorical or binary variables. This makes accurate biological measurements such as GDF15 levels in some cases a much better estimator of biological changes than phenotypic data. However, we think a lot of the prediction power of GDF15 is related to its increased expression in serious diseases such as cancer and cardiovascular diseases. It is also clear from Supplementary Figure 4, where the baseline + GDF15 is considerably better than age+sex+GDF15 that lifestyle and disease variables in the baseline capture something that GDF15 alone does not.

We examined prediction using other single proteins. The proteins most likely to be effective are those strongly associated with mortality. Therefore, we tried the top-ten proteins associated with five-year all-cause mortality. Prediction performance was consistent with the association strength, with the highest association yielding the best predictors. We have now included in the revised manuscript predictions using three of the other top proteins in addition to GDF15 as outlined in Supplementary Figure 5a and the text below has been added to the Results:

To see if other single proteins were good mortality predictors, we also tried models using age, sex, and WAP four-disulfide core domain protein 2 (WFDC2), Thrombospondin-2 (THBS2), or Anthrax toxin receptor 2 (ANTXR2). These were the proteins in addition to GDF15 with the strongest association with five-year mortality and were all useful in predicting mortality. However, the GDF15 model remains the only single protein model to surpass the baseline model in prediction performance (Supplementary Figure 5a). Therefore, GDF15 cannot easily be swapped for any single candidate protein, but a combination of proteins can make up for performance loss from excluding GDF15.

We also looked at how these single protein predictors performed for different causes of death, included the results in Supplementary Figure 5b, and added the text below to the Results:

Single protein predictor performance was in accordance with the univariate associations. Supplementary Figure 5b shows that ANTXR2 was the top protein in predicting nervous system-related deaths. WFDC2 was best at predicting respiratory system related deaths, while GDF15 was best at predicting neoplasm and other deaths. GDF15 and WFDC2 were best at predicting circulatory system deaths.

Supplementary Figure 5: AUC for models using single proteins. a) Prediction for different time points, b) five-year prediction for different causes of death.

11. What would a sensitivity analysis excluding GDF15 from all the protein model show?

This is an interesting question. We created new protein models where GDF15 was not one of the candidate proteins and found that excluding GDF15 had little effect. To include these results, we changed Supplementary Figure 4 to include protein models where GDF15 is excluded and added the following text to the Results:

GDF15 is a powerful predictor on its own and as a part of the protein model. To examine what a protein model without GDF15 could do, we created a new protein model where we excluded GDF15 (Supplementary Figure 4). There was no significant difference in the AUC between the protein model and the new protein model excluding GDF15.

Supplementary Figure 4: AUC at different time points for models with and without baseline features and with and without GDF15. a) All participants, b) participants older than 60.

12. Using cox survival analysis as sensitivity analysis instead of logistic regression for univariate associations could be helpful. Time to death could be used instead of 5 years cutoff.

We tried both approaches, and agree that Cox survival analysis may seem the most natural choice. Logistic regression was chosen since it better matched the methods we used for our prediction model, and it was a simple way to see if there was a difference in the protein profiles for short- and long-term prediction. Using Cox proportional hazards models, the proportional hazards assumption was violated in most cases, and measures to address that would, in our opinion, take away from the interpretability. In light of the reviewer's comment, we have included the significant Cox associations with all-cause mortality and all significant five-year mortality associations in Supplementary Tables 9-11. The following text has been added to the Results:

We also examined associations using the Cox proportional hazards model with similar results. Other single protein associations are in Supplementary Table 9 - 11.

13. It is interesting to note that top single protein association with Nervous system (Supplementary Table 6) are having negative Beta, while most of other systems top proteins have positive Beta value. What are the possible explanations? Also, what are the possible explanations for little success in capturing nervous system associated death. Disease progression time as well as severity of the diseases varies. How much does it influence the prediction modeling?

Under the very strict Bonferroni correction, only seven proteins were found to be significantly associated with nervous system related deaths within five years, only one positively, and one of them was not associated with all-cause mortality. The top protein for most of the other systems, GDF15, was not associated with nervous-system deaths even without multiple-testing correction of the p-value. It is also true that all other systems have considerably more positive protein associations than negative. We don't have a definite explanation for the different protein profile in nervous system related mortality. It appears as other systems' deaths have common factors that are not seen for the nervous system deaths.

A part of the reason the protein model cannot predict nervous system deaths well might be because we have relatively few nervous system deaths in our dataset and because their protein profile is considerably different from the other causes of death. Other systems with few cases, like the respiratory system, probably suffer less because the top proteins associated with death are also the top all-cause mortality proteins. This is not the case for the nervous system.

We have extended the discussion on this protein profile discrepancy in the Results in the revised manuscript as follows:

When univariate associations were examined separately for different causes of death (Supplementary Table 5), the top two proteins for all-cause mortality, GDF15 and WFDC2, were among the top five proteins for all the death categories except for the nervous system category. The nervous system category had a different protein profile from the other categories, with the top four proteins having negative associations with death and GDF15 far from being significantly associated with death. This difference is probably partly responsible for the slight improvement of the protein model over the age and sex model in predicting nervous system deaths, as depicted in Figure 3. All top five proteins for neoplasms were also in the top ten for all-cause mortality. Because neoplasms are the most common cause of death in our data, it is not surprising that all-cause mortality and neoplasm mortality have similar protein profiles.

We have also added a paragraph on this to the discussion:

Examining protein profiles of different disease categories of death revealed that most major systems, except the nervous system, have similar profiles. It is fascinating to find that GDF15, which has the biggest effect by a large margin in most categories, is not significantly associated with nervous system deaths. Despite these differences, the top associations with nervous system deaths were all significantly associated with all-cause mortality and in the same directions. From this, it seems that deaths connected to the nervous system lack factors common to other systems.

This comment from the reviewer also prompted us to include the number of positively and negatively associated proteins in the Results:

In a univariate association corrected for age and sex, 1,364 protein measurements out of 4,905 were significantly associated with death within five years after a Bonferroni correction, 1292 positively and 72 negatively. For participants older than 60 years, the number of associated protein measurements was 1,068, 1002 associated positively, and 66 negatively.

Disease progression time and severity varies for most diseases. To evaluate how this influences the prediction we would need to have a more complete medical history for all participants. This information we do not have and even if we did, evaluating this would still be hard. We can only speculate that short-term predictions are generally easier than long-term since those who die soon after plasma collection are likely to have severe and fast progressing diseases, which are likely to influence the proteome a lot. In the case of the proteomics data it is also possible that some of the prediction performance comes from detecting treatments, such as medication use and major surgeries, which are likely to influence the proteome.

14. What are the possible explanations of PRS not having a drastic effect in prediction modeling?

We speculate that the reason for the little effect of including PRS in our risk models is because PRS is a long-term (lifetime) predictor that does not change through the course of life. In our relatively short time predictions, the genetic effects are probably overshadowed by effects of lifestyle and age. We have added a ten-year prediction to Supplementary Figure 17, where the PRS seems to improve the prediction slightly, supporting the theory that PRS is more useful for longer-term prediction. We also extended the paragraph discussing PRS in the Methods as follows:

Polygenic risk scores (PRSs) for cancer, hypertension, stroke, CAD, Alzheimer's, attention deficit hyperactivity disorder (ADHD), Parkinson's, educational attainment, depression, bipolar disorder, BMI, schizophrenia, IQ, autism, and anorexia were tested as features in the all-cause mortality risk predictor. They did not improve the baseline prediction for five-year mortality and only improved the ten-year prediction slightly (Supplementary Figure 17). Based on this, we think that including PRSs will have more value in longer-term predictions. Due to their small effect, the PRSs were not used in any of our prediction models.

Supplementary Figure 17: Ten-fold CV AUC for models with and without PRS. The yellow centre line represents the median, the box limits represent the upper and lower quartiles, the whiskers represent the 1.5x interquartile range, the circles are data points outside the whisker range, and the blue dots represent the mean values.

Minor comments

1. Why was L1 penalty selection of 219 proteins for 13 years mentioned in line 87 than 192 for 15 years?

The number of proteins for 13 years was mentioned because the model for death within 13 years includes more proteins than the models for the other years. We modified the sentence to include the number of proteins for 15 years as well.

The L1²⁵ penalty reduced the model to 81 protein measurements for prediction of death within one year and 192 for death within 15 years, but the biggest model, which was for prediction of death within 13 years, used 219 protein measurements.

2. Line 112 mentions supplementary Figure 1. Instead of supplementary Figure 2. Supplementary figures number should be redone as there are changes at multiple places.

We have gone carefully through this in the revised manuscript and made sure there are no errors.

3. Line 167 "to and age and sex" please check

We have changed this to "to age and sex"

4. Lines 248-249 please rephrase the sentences

We changed the entire paragraph to:

The largest multi-biomarker all-cause mortality study we know of is the metabolomics study by Deelen et al.⁸ They use a dataset of 44,168 individuals, which is far greater than ours, but in our dataset, we have more deaths during the study period and a larger set of biomarkers measured. We have not seen proteomics all-cause mortality studies with as many participants as we have. Therefore, we believe our study is the largest to date.

Extensive proteomic data used in this study will definitely provide important contribution in understanding basis of complex phenotypes of aging as well as age associated diseases.

Reviewer #2 (Remarks to the Author):

The authors set out to predict mortality based on levels of thousands of proteins in plasma from more than 20,000 individuals age 18-101. They develop various models using demographic and other lifestyle assessments in combination with proteomics to develop risk assessments for various times into the future. Overall, the size of the cohort and the number of proteins measured are most impressive and an incredibly valuable treasure trove and the reported results are clear.

What this reviewer missed was an interpretation of the findings, biological significance of the observed top predictor proteins, a discussion of the potential utility of the findings, and a comparison with all the other mortality predictors out there. After reading the article I was not clear what the next steps were and why the authors carried out the study. The final statement in the abstract " Our results show that the plasma proteome can be used to estimate general health and the risk of death" is rather non-specific and, as the authors discuss in the introduction, not novel.

We thank the reviewer for the helpful comments. We have tried to extend our interpretations of the findings, add some analysis of biological contributions and comparison to other protein predictors. (See detailed answer to remark 3 from reviewer 2). Our focus was on using the proteins as biomarkers. Therefore we have not gone deeply into biology, but we felt that the reviewer was correct and some attempt in that direction would add considerably to the paper.

What we wanted to do was to predict the probability of death using the proteome. Thereby showing the potential of the proteome as a powerful prognosis and health estimator. Even if our predictor is not likely to be clinically significant, we were able to show that the proteome can be used reliably for such predictions. Some studies have considered the connections between the proteome and all-cause mortality before but never for so many proteins and so many people. We think that adds considerably to the knowledge of how well the proteome reflects an individual's health. As we talk about in the discussion, the next steps we envision involve more specific questions. Examining the connection between the proteome and specific causes of death and specific proteins and death. A deep dive into biology could help reveal causal relationships and possibly identify potential drugs or drug targets. The possibilities are endless, but we think we have shown an interesting potential use for the proteome in health care, although not completely novel, never shown as clearly in this context before.

Major questions:

1) The study combines multiple cohorts and it appears they have not been merged or normalized. In similar studies using the Somalogic platform, a simple PCA revealed the origin of samples based on cohort or center of collection before normalization (some studies did not normalize and analyzing their data reveals this major flaw). This needs to be addressed. It is not stated how samples were collected (EDTA, heparin, citrate) but I assume they were of the same type?

Even though our data was sampled for many different projects it was sampled at only two sites. One site was focused on cancer research for many different types of cancer, while the other sampled blood for 41 different projects exploring various phenotypes. The blood was sampled the same way

for all the different projects at both sites. We have added a paragraph in the revised manuscript Methods describing the plasma sample collection process.

Blood was collected in EDTA tubes. The tubes were inverted 4–5 times before being centrifuged for 10 min at 3,000 g at 4 °C. Plasma samples were frozen in aliquots at –80 °C. Plasma aliquots were kept away from light while they were allowed to thaw on ice. The aliquots were mixed by inverting the tubes three times and then centrifuged for 10 min at 3,220 g at 4 °C before measurement.

There are other possible sources of errors in the protein measuring process, e.g., differences between plates and analysed batches. Principal component analysis of the proteomics data revealed the first principal components to be most correlated to lifestyle factors, such as age, sex, and bmi. Other in house experiments also revealed that major errors could be introduced in the normalization process done at SomaLogic, which is why we did not use it. Even though we could not correct directly for these possible errors some quality control was applied as described in the methods and repeated measurements of the same samples were used to exclude probes that were not robust between measurements.

We experimented with correcting the proteins for site and sampling year before training the model. As we can see in Review Figure 1 below, there was no noticeable performance difference before and after correction. To address the same question, we trained a predictor in the ICP data and predicted in the VSP1 data, where the main results, presented in Review Figure 2, were the same as when the datasets were mixed. We also tried univariate analysis for five-year mortality where we included site and sampling year as covariates, that did not change which proteins were in the top ten and only altered the betas slightly.

Since the mentioned factors, i.e., site, sample age, and other possible differences in sample treatment, did not have big effects on the measurements or predictions, we chose not to correct for them. Our experiments also showed that normalizing the data, i.e., force to a normal distribution, did not help the prediction, just using the log-transform was better. This simple pre-processing makes it possible to apply our predictor to any independent measurements from the SOMAscan platform. (Using another dataset using the same SOMAscan platform, we could predict age with a predictor trained in our data with comparable accuracy in both datasets. We did not try to predict mortality because we had no follow-up information. We could not use the results from this experiment since we do not have permission to use the data in publications.)

Review Figure 1: Effects of correcting protein levels for sampling site and sampling year or batch compared to no correction and the baseline model.

Review Figure 2: Prediction with predictor developed in ICP data and tested in VSP1 data.

2) Related to 1) can the authors build a model in one cohort and predict mortality with similar accuracy and using the same predictors in an independent cohort?

The data we have available are sampled at different times for different purposes, but the protein levels are measured at the same time on the same assays making it hard to call the datasets completely independent. As a step in the direction of using a separate cohort, we trained on the ICP data and tested on VSP1 and achieved similar improvement by the protein model over the baseline model as in our original experiments, Review Figure 2.

We also use our original prediction model trained on 70% of the ICP+VSP1 dataset to predict in the DHS+VSP2 dataset. The predicted risks in the living participants and dead participants are in Supplementary Figure 13. To make the comparison between models clearer, we added a comparison of C-indexes to the text in the Results.

The baseline had a C-index of 0.900, while the protein model had 0.938. In this case, we use C-index rather than AUC because of extensive censoring from limited follow-up in this dataset.

3) I had a hard time reading this article as it basically recites all the numbers and p-values which are, or which should be, in figures - hence we use graphics for scientific presentations. It would be more important to tell the reader what the results mean:

We agree that there were too many recited numbers in some parts of the results, especially the chapter "Prediction performance at different time points." We have addressed this by removing recited numbers connected to other than five-year prediction. We agree that they were not necessary since the general result is shown in figures, and the numbers can be looked up in supplementary tables. We also think that the inclusion of pathway analysis, comparison to other predictors, and more experiments suggested by the reviewers have made the paper more readable.

Why are these specific proteins selected into the predictor?

The specific proteins included in the predictor are chosen because they generate the best predictor. We did not consider the biology of the proteins at all but let the model select the proteins that gave the best prediction. In the revised manuscript we emphasised this point by adding the following sentence to the Discussion:

The proteins were selected by the model without any consideration of their biology.

Why are there different ones for the various diseases?

Most causes of death had similar top proteins, with GDF15 clearly being the most important protein. This we felt suggested something common to most causes, which is what we would expect an all-cause mortality predictor to use. Different proteins are probably both caused by the different progress of various diseases and in the case of few deaths chance could affect the order of proteins very similar effect sizes considerably. We do not discuss this much since we do not go deeply into the different causes and find it more interesting how universal some of the proteins are. In the case of the nervous system, which is strikingly different from the rest we have added some discussion on the matter (See answer to remark 13 from reviewer 1).

Are they in specific pathways or does it mean anything biologically?

We did simple pathway analysis and added some discussion on GDF15. (See answer to remarks 1 and 2 from reviewer 1)

Have other studies identified similar biology as discussed in the introduction, what are the overlapping, consistent proteins, etc.?

As mentioned in previous answers, we have included discussion on previous research on GDF15 (see answer to remark 2 from reviewer 1) and discussed previous results in our discussion, both with the comparison to other protein predictors (see answer to remark 1 from reviewer 1) as well as the following paragraph in the Discussion:

We identified more than a thousand proteins associated with all-cause mortality, confirming many that have been identified before^{12,13,18} and identifying new ones. It is interesting to note that less than 1% of the associated proteins were negatively associated with death. Previous studies have shown that most plasma protein levels rise with age, especially after 60²⁰. We speculate that these associations have a common root and are likely to be either caused by the body starting to fail or as a response to counteract the failing.

References

1. Tanaka, T. *et al.* Plasma proteomic biomarker signature of age predicts health and life span. *eLife* **9**, e61073 (2020).
2. Sathyan, S. *et al.* Plasma proteomic profile of age, health span, and all-cause mortality in older adults. *Aging Cell* **19**, e13250 (2020).
3. Orwoll, E. S. *et al.* High-throughput serum proteomics for the identification of protein biomarkers of mortality in older men. *Aging Cell* **17**, e12717 (2018).
4. Tanaka, T. *et al.* Plasma proteomic signature of age in healthy humans. *Aging Cell* **17**, (2018).
5. Menni, C. *et al.* Circulating Proteomic Signatures of Chronological Age. *J Gerontol A Biol Sci Med Sci* **70**, 809–816 (2015).
6. Ho, J. E. *et al.* Protein biomarkers of cardiovascular disease and mortality in the community. *Journal of the American Heart Association* **7**, (2018).
7. Deelen, J. *et al.* A metabolic profile of all-cause mortality risk identified in an observational study of 44,168 individuals. *Nat Commun* **10**, 3346 (2019).

Reviewers' Comments:

Reviewer #1:

Remarks to the Author:

Authors have worked extensively in answering the comments as well as made changes in the manuscript. Altogether the manuscript has improved in its format. I still have few queries which basically focusses on the model adopted for this study and trying to make it more 'clinically' relevant.

1. Why not focus on the group with the most substantial risk? Is the approach adopted in this study, of predicting deaths in years and then looking at the success rate, best approach to be used? Author finds 5% at highest risk for death in 60-80 years old and reported 88% death within ten years. But if we look at another angle, 6,222(61.4%) out of 10,136 participants died in ~10 years of follow-up(60+group). That means even if we take a random sample there is a probability of 61% deaths to happen in 10 years. I think in that case major finding in this manuscript is observation of lowest risk group in which only 1percent participant succumbed to death.
2. Chronological age and biological risk group? Authors identified 5% with an 88% probability of death within 10 years and 5% group with less probability of dying within 10 years. What was the mean age and SD of these two groups? Were people close to 80 years more prevalent in 5% group with higher mortality and vice versa with healthy group. Though it is 20-year window, the mean age might be crucial as age itself is a risk factor for mortality
3. Eruption over overexpressed proteins ads risk for death? It is very intriguing to see just 72 out of 1364 SOMAmers (5%) negatively associated with death within five years in univariate model. Earlier studies have discussed about equidistribution of positively and negatively associated proteins with chronological age. Does author feel that an eruption of overexpressed proteins is leading to death? It will be nice if author could explain based on biological context.
4. Are different models using different participants? Author predicted 5% (124 participants) at highest risk independently using different models. What was the overlap of participants seen in all these models? Will the shared participant pool be at greater risk for death?
5. Pooled shared proteins and risk of death? What was the reason for drastic difference in the "cumulative overlap of previous years proteins" in the Boruta and Lasso model? For example, in year-15, 10 out of 192(0.05%) had cumulative overlap in Boruta and 135 out of 454(29%) overlapped in Lasso. Is this trend in cumulative overlap expected for a common outcome(death) measured for ascending years (1-15 years)? How does this explain in context of biology? Is this pool of shared proteins playing important role in Aging?
6. GDF15 and time to death? Present models of GDF15 and proteins are built upon actual chorological age + gender. Is there a possibility of coming up with a model ignoring actual chronological age and just focusing on time to death using only proteomic profile.

Minor comments

1. Please check line no 117 and 118, IDI values seem to be slightly different from the previous version of the manuscript. What was the reason for this change?

Reviewer #2:

Remarks to the Author:

I'm impressed by the effort and seriousness of the response to my and the other reviewer's comments and i have no further comments or concerns and recommend publication

We thank the reviewers and the editor for their comments. Below we respond to each comment point-by-point. In addition, we have made some additional improvements as described below.

We substituted the exercise test phenotype from maximum exercise time to VO2 max. This required changes to Table 2 and the corresponding text in the manuscript but did not change the association between exercise test performance and mortality risk. We did this because it was brought to our attention that VO2 max is more commonly used to measure endurance fitness. We also modified the text in the Methods section and updated Supplementary Table 13 to make some facts regarding our baseline features clearer. The order of some paragraphs in the Methods was changed to better reflect the order in the Results. Also, we found that we had not included stroke cases in patients who also experienced an MI. The inclusion of these cases resulted in minor changes to the baseline that in no way affects our conclusions but required an update on some figures and numbers. Two more authors were found to have contributed and were consequently added. Finally, we fixed a few grammatical errors and reworded and added a few sentences for clarity.

The reviewers' comments are included as they were written, but in bold letters, our answers are in italics, text from the manuscript is in italics and underlined. All significant changes are highlighted in the revised manuscript.

Reviewers' comments:

Reviewer #1 (Remarks to the Author):

Authors have worked extensively in answering the comments as well as made changes in the manuscript. Altogether the manuscript has improved in its format. I still have few queries which basically focusses on the model adopted for this study and trying to make it more 'clinically' relevant.

1. Why not focus on the group with the most substantial risk? Is the approach adopted in this study, of predicting deaths in years and then looking at the success rate, best approach to be used? Author finds 5% at highest risk for death in 60-80 years old and reported 88% death within ten years. But if we look at another angle, 6,222(61.4%) out of 10,136 participants died in ~10 years of follow-up(60+group). That means even if we take a random sample there is a probability of 61% deaths to happen in 10 years. I think in that case major finding in this manuscript is observation of lowest risk group in which only 1percent participant succumbed to death.

We chose to examine the 60-80-years group rather than the whole 60+ group because almost every 80+ participant is likely to die within ten years. We think that our predictor has the clearest advantage and is most likely to be clinically useful in a group with high variability in mortality risk. In the manuscript, we mention that in the 60-80-years-old group, 28.2% die within ten years, making 88% considerably higher than in a random sample. The low-risk group with a 1% probability of dying within ten years includes one death in the group vs. seven deaths in a similar group identified with the baseline model. While that is an impressively low risk and potentially useful, we fear that this result is more likely to be based on chance and not as reliable as the difference between high-risk groups. In Supplementary Figure 8, we look at survival in high- and low-risk groups of 80+ year olds. There, 96% die within ten years among the 20% at the highest risk group identified by the baseline model, while 99% die within ten years in the protein model identified group. In this group, long-term differences between 20% at lowest risk identified by the different models are more interesting; 45% die within ten years in the group identified by the protein model while 56% die within ten years in the baseline model identified group.

2. Chronological age and biological risk group? Authors identified 5% with an 88% probability of death within 10 years and 5% group with less probability of dying within 10 years. What was the mean age and SD of these two groups? Were people close to 80 years more prevalent in 5% group with higher mortality and vice versa with healthy group. Though it is 20-year window, the mean age might be crucial as age itself is a risk factor for mortality

As the reviewer mentions, age is an important risk factor for mortality. Therefore it is expected that the high-risk groups are older on average than the low-risk groups. The mean age in the high-risk group identified by the protein model is 74.0 (sd. 4.9) and 77.0 (sd. 2.5) in the baseline model identified group. The protein model low-risk group has an average age of 62.7 (sd. 2.4), while the baseline low-risk group has a mean age of 61.8 (sd. 1.4). The average age in the whole group is 70.1 (sd. 5.5). The distribution of age in these groups is demonstrated in Review Figure 1. There we see that the age in the high-risk groups is usually more widespread than in the low-risk groups and that the protein model high-risk group includes patients of almost every age in the 20-year window.

We feel that a comparison to a model that only uses age and sex clearly shows that age is an important predictor. Still, the protein prediction model uses more than just age to classify and is superior to just using age. We found this age comparison interesting and decided to include it in the revised manuscript Results in the following paragraphs.

We looked at Kaplan-Meier survival curves for participants in the ICP+VSP1 test set between 60 and 80 years old to reduce the effect of age. That included 2,488 participants with mean age 70.1 (sd. 5.5), of whom 1,312 (53.1%) died during the study period, 305 (12.6%) within five years, and 701 (28.2%) within ten years from sample collection. The curves were plotted separately for the four prediction models. By splitting the Kaplan-Meier curves by quantiles of predicted ten-year risk, the proteins' discriminative power becomes evident (Figure 2, Supplementary Figure 7). Of the 5% (124 participants) predicted at the highest risk by the age and sex, baseline, GDF15, and protein model, 25%, 40%, 55%, and 67% died within five years, and 56%, 65%, 74%, and 88% within ten years. Of the 5% (125 participants) predicted at the lowest risk by each model, 8%, 5%, 8%, and 1% died within ten years.

The protein model 5% high-risk group is younger and of more varied age (mean 74.0, sd. 4.9) than the baseline model group (mean 76.9, sd. 2.6). Likewise, the protein model 5% low-risk group is older and of more varied age (mean 62.7, sd. 2.4) than the baseline model group (mean 61.9, sd. 1.3).

Additionally, we added the following text to the Discussion in the revised manuscript.

The protein model also relied much less on age in separating high- and low-risk groups than the baseline model. The difference in age between the groups was less for the protein model than the baseline model and the variance in age in each group higher.

Review Figure 1: Age distribution in high- and low-risk groups of 60-80 year old participants as identified by the different models.

3. Eruption over overexpressed proteins ads risk for death? It is very intriguing to see just 72 out of 1364 SOMAmers (5%) negatively associated with death within five years in univariate model. Earlier studies have discussed about equidistribution of positively and negatively associated proteins with chronological age. Does author feel that an eruption of overexpressed proteins is leading to death? It will be nice if author could explain based on biological context.

Studies that use SomaScan platforms to measure plasma protein levels and examine how they associate with age have found that more of the measured proteins are positively associated with age than negatively¹⁻⁴. The difference between the numbers of negatively and positively associated proteins is not as extreme as we have found for mortality. Other studies of protein level associations with mortality have found more positively associated proteins than negatively^{5,6}. We think rising protein levels with age and the association between high protein levels and mortality are probably related and connected to the body starting to fail or defensive responses to counteract the failing. In most cases, we believe, higher protein levels are caused by conditions that lead to death rather than the other way around.

One reason for the difference in the number of positive and negative associations could be that we measure a limited set of plasma proteins which might be more correlated than perfectly randomly selected proteins. Another reason may be that there are factors that influence total protein levels in the plasma, resulting in multiple protein levels going up or down. Some of these factors are likely correlated with mortality risk resulting in most of the proteins associating with mortality in the same direction.

4. Are different models using different participants? Author predicted 5% (124 participants) at highest risk independently using different models. What was the overlap of participants seen in all these models? Will the shared participant pool be at greater risk for death?

The intersection of high-risk groups identified by all four models includes 14 participants, of whom 13(93%) die within ten years. This probability of death is higher than that of the 124 participants in the high-risk group identified by the protein model. However, of the 24 participants predicted at the 1% highest risk by the protein model, 23 (96%) die within ten years, suggesting that the overlap of the four predictors is not a promising approach to improve the protein model's predictions.

5. Pooled shared proteins and risk of death? What was the reason for drastic difference in the "cumulative overlap of previous years proteins" in the Boruta and Lasso model? For example, in year-15, 10 out of 192(0.05%) had cumulative overlap in Boruta and 135 out of 454(29%) overlapped in Lasso. Is this trend in cumulative overlap expected for a common outcome(death) measured for ascending years(1-15 years)? How does this explain in context of biology? Is this pool of shared proteins playing important role in Aging?

The difference between the number of cumulatively selected proteins by Lasso and Boruta is both due to the different sets the proteins were selected from and how they are selected. (Boruta and Lasso are switched in the question, Lasso selected 10 proteins in every model and Boruta 135).

The set of proteins selected by Boruta is not corrected for age, and the aim of the method is to select all proteins relevant to the prediction rather than just a sufficient set. It should not affect the selection of one protein if a related protein has already been chosen. Therefore, most of the proteins selected for the prediction of death within one year are also chosen for longer periods. The difference between years should mostly be because of the randomness of the method or proteins that are important for short-term prediction not being as important when short-term and long-term cases do not need to be separated.

The Lasso selects proteins from the Boruta set and can therefore never select more proteins than Boruta. Lasso selects the smallest set for good prediction. Consequently, it is very likely to pick only one of many highly correlated proteins. When we change the endpoint to death within more years, Lasso could easily select a different protein from a set of highly correlated proteins, especially if the other proteins reflect long-term health better. We would expect the set of proteins that the Lasso always chooses to be better predictors of mortality than correlated proteins. The Lasso model also includes age. Therefore age might be selected as a feature rather than some proteins highly correlated with normal ageing.

The pool of selected proteins should be a selection of proteins that are important for mortality prediction. Among the ten proteins selected by the Lasso for mortality prediction of all participants are GDF15, WFDC2, MMP12, and ANTRXR2. These proteins are among the proteins we have found to be most strongly associated with mortality. All except one associate with age in a study by Sathyan et al. ¹. Three of them among the top 20, and all are associated with age in our data. All but two of the Boruta pool are associated with age in our data. Therefore, we believe these pools of shared proteins are important both in age and mortality prediction.

To summarize, Lasso selects the smallest set of proteins and corrects for age, while Boruta selects all-relevant proteins without any correction. The most interesting conclusion from this, in our opinion, is that while many proteins are associated with mortality and can be useful predictors, few are essential; most can be easily swapped for other proteins, still yielding good predictions.

6. GDF15 and time to death? Present models of GDF15 and proteins are built upon actual chronological age + gender. Is there a possibility of coming up with a model ignoring actual chronological age and just focusing on time to death using only protec profile.

We also created models using only the protein levels. The comparisons to models including age and sex are in Review Figure 1. When we let the model select multiple proteins, including age and sex was not very important, especially for short-term predictions. For long-term predictions, the model including age and sex is slightly better than the one without. Excluding age and sex generally results in more proteins being used for the prediction than when age and sex are included. For the five-year prediction, the model without age and sex used five more proteins than the one including age and sex, and seven more for the ten-year prediction. When the only protein used is GDF15, age and sex are an important part of the models, especially for long-term predictions.

Since age and sex are usually easily obtainable features, we did not see much advantage in excluding them. However, it is interesting to see that the proteome accounts almost perfectly for chronological age effects. Therefore we have added the following text to the Results in the revised manuscript.

Excluding age and sex from the protein model reduced prediction performance slightly, especially for long-term predictions, but age and sex are an essential part of the GDF15 model. Since age and sex are easily obtainable features, we saw no advantage in excluding them from the models.

Review Figure 2: AUC at different time points for protein mortality risk prediction models with and without age and sex.

Minor comments

1. Please check line no 117 and 118, IDI values seem to be slightly different from the previous version of the manuscript. What was the reason for this change?

During the first revision we updated the baseline predictor to include information on history of stroke. This slightly altered the performance of the baseline predictor. Since the IDI is the improvement over the baseline predictions, the IDI for all predictions changed. We did, however, not include all stroke cases in that version but they have now been included, which resulted in updated numbers in the current form of the revised manuscript.

Reviewer #2 (Remarks to the Author):

I'm impressed by the effort and seriousness of the response to my and the other reviewer's comments and i have no further comments or concerns and recommend publication

References

1. Sathyan, S. *et al.* Plasma proteomic profile of age, health span, and all-cause mortality in older adults. *Aging Cell* **19**, e13250 (2020).
2. Lehallier, B. *et al.* Undulating changes in human plasma proteome profiles across the lifespan. *Nature Medicine* **25**, 1843–1850 (2019).
3. Tanaka, T. *et al.* Plasma proteomic signature of age in healthy humans. *Aging Cell* **17**, (2018).
4. Tanaka, T. *et al.* Plasma proteomic biomarker signature of age predicts health and life span. *eLife* **9**, e61073 (2020).
5. Ho, J. E. *et al.* Protein biomarkers of cardiovascular disease and mortality in the community. *Journal of the American Heart Association* **7**, (2018).
6. Orwoll, E. S. *et al.* High-throughput serum proteomics for the identification of protein biomarkers of mortality in older men. *Aging Cell* **17**, e12717 (2018).

Reviewers' Comments:

Reviewer #1:

Remarks to the Author:

This is an extensive dataset(25k individuals) which could be of great resource in future.

The manuscript has improved in a great way from initial submission. Authors have addressed all of the queries.